# Osteoclast-associated receptor blockade prevents articular cartilage destruction via chondrocyte apoptosis regulation

Doo Ri Park[1,2], Jihee Kim [1,2], Gyeong Min Kim[1,2], Haeseung Lee [1], Minhee Kim[1,2], Donghyun Hwang [3], Hana Lee[3], Han-Sung Kim[3], Wankyu Kim [1], Min Chan Park[4], Hyunbo Shim[1] & Soo Young Lee [1,2✉]

Osteoarthritis (OA), primarily characterized by articular cartilage destruction, is the most common form of age-related degenerative whole-joint disease. No disease-modifying treatments for OA are currently available. Although OA is primarily characterized by cartilage destruction, our understanding of the processes controlling OA progression is poor. Here, we report the association of OA with increased levels of osteoclast-associated receptor (OSCAR), an immunoglobulin-like collagen-recognition receptor. In mice, OSCAR deletion abrogates OA manifestations, such as articular cartilage destruction, subchondral bone sclerosis, and hyaline cartilage loss. These effects are a result of decreased chondrocyte apoptosis, which is caused by the tumor necrosis factor-related apoptosis-inducing ligand (TRAIL) in induced OA. Treatments with human OSCAR-Fc fusion protein attenuates OA pathogenesis caused by experimental OA. Thus, this work highlights the function of OSCAR as a catabolic regulator of OA pathogenesis, indicating that OSCAR blockade is a potential therapy for OA.

[1] Department of Life Science, Ewha Womans University, Seoul 03760, South Korea. [2] The Research Center for Cellular Homeostasis, Ewha Womans University, Seoul 03760, South Korea. [3] Department of Biomedical Engineering, Yonsei University, Wonju 26493, South Korea. [4] Division of Rheumatology, Department of Internal Medicine, Yonsei University College of Medicine, Seoul 06273, South Korea. ✉email: leesy@ewha.ac.kr

Osteoarthritis (OA), the most common degenerative joint disease afflicting the knee joints, is caused by many risk factors[1,2]. OA is primarily characterized by chronic inflammation and collagen degradation in the articular cartilage, leading to progressive irreversible dysfunction[3–6]. Although studies have identified elements that are involved in the development of OA, the importance of collagen recognition for cartilage damage and regeneration is poorly understood.

Articular cartilage includes chondrocytes specifically responsible for cartilage homeostasis and OA pathogenesis[7–9]. OA is associated with the degradation of extracellular matrix molecules such as collagen and aggrecan (ACAN) by the reduced ability of chondrocytes to generate articular cartilage extracellular matrix and the action of catabolic matrix-degrading enzymes[10–12]. OA pathogenesis is also related to chondrocyte apoptosis and reduction of tissue cellularity. Apoptotic chondrocytes are more frequently found in cartilage lesions than in healthy tissues and exhibit a proteoglycan deficiency[13–15]. However, the high rate of cell death in cartilage is generally a rapid process, and the functional relationship between OA and chondrocyte apoptosis is difficult to assess.

Osteoclast-associated receptor (OSCAR) is an activating receptor for collagen that co-stimulates osteoclast differentiation[16]. OSCAR is an immunoglobulin (Ig)-like activating receptor of the leukocyte receptor complex that is specifically expressed on pre-osteoclasts. In osteoclasts, OSCAR is mediated by the immunoreceptor tyrosine-based activation motif adaptor protein FcRγ and co-stimulates osteoclasts cultured in an extracellular matrix or collagen[17,18]. OSCAR contains two immunoglobulin-like domains, D1 and D2. Recently D2, but not D1, was shown to be critical for collagen binding[19,20]. In human rheumatoid arthritis (RA), OSCAR is induced in monocytes, facilitating their differentiation into osteoclasts and bone resorption[21]. OSCAR is also expressed on the surrounding cells of synovial microvessels and is upregulated in peripheral blood monocytes in relation to disease activity[21–23]. In a recent study, the OSCAR–collagen interaction induces an activating signal in monocyte-derived DCs, which is relevant to RA pathogenesis[24]. However, limited information is available on the effects of OSCAR in articular cartilage. Moreover, there appears to be no experimental evidence with respect to OSCAR function in osteoarthritic cartilage.

In this study, we examined whether OSCAR, a collagen-recognition receptor, contributes to cartilage destruction during OA. We found that the inhibition of OSCAR activity attenuated the pathological changes of OA in the subchondral bone and reduced the degeneration of articular cartilage in OA-induced models. Besides, intra-articular (IA) injection of human OSCAR-Fc protein in vivo blocks OA development. In this study, we determined that blocking OSCAR may be the basis of a potential disease-modifying therapy in OA.

## Results

**OSCAR is increased in OA chondrocytes.** OSCAR is upregulated in a range of myeloid cells, including osteoclasts[17,18]. We detected OSCAR in the messenger RNA (mRNA) and protein of chondrocytes in wild-type (WT) mice (Fig. 1a) but not in OSCAR knockout (Oscar−/−) mice[16], suggesting that OSCAR is also expressed in nonmyeloid cells. We further investigated OSCAR expression in chondrocytes of cartilage in an experimental OA model in mice. Immunostaining revealed that the OSCAR protein level was markedly elevated in chondrocytes in OA induced by surgery performed for the destabilization of the medial meniscus (DMM)[25] (Fig. 1b, Supplementary Fig. 1a, b). Consistent with increased protein levels, we found that Oscar mRNA expression increased in mice with OA (Fig. 1b, Supplementary Fig. 1c).

Similarly, OSCAR was significantly increased in OA-affected, damaged regions of human articular cartilage compared with undamaged regions (Intact) of arthritic cartilage (Fig. 1c). Furthermore, OSCAR was expressed in the early stages of OA, and it was further induced during later stages (Supplementary Fig. 1a, b). However, the expression of matrix-degrading enzymes, including matrix metalloproteinases-3 and -13 (MMP3 and MMP13), increased at 4 weeks following surgery (Supplementary Fig. 1d–g). These findings suggest that OA-associated catabolic events in mice and humans upregulate OSCAR expression in the early stages of OA.

**Genetic deletion of Oscar reduces OA pathogenesis.** To explore the possible association of OSCAR with OA pathogenesis initiated by articular cartilage degradation at the articular surface, we first induced experimental OA with collagenase[26,27], which is an enzymatic degradation model emulating the early stages of articular cartilage damage in Oscar−/− mice. Important hallmarks of OA pathology including articular cartilage destruction, subchondral bone sclerosis, and hyaline cartilage loss, were substantially inhibited in Oscar−/− mice compared with that in WT mice (Supplementary Fig. 2a–d). Moreover, the expression of matrix-degrading enzymes, including Mmp3, Mmp13, and Adamts5, were significantly decreased (Supplementary Fig. 2e), whereas the transcript levels of type II collagen (Col2a1) and Acan were increased in OA chondrocytes from Oscar−/− mice (Supplementary Fig. 2f). Similarly, immunohistochemical analysis revealed decreased expression of MMP3, MMP13, and ADAMTS5, and upregulation of aggrecan and COL2A1 in OA chondrocytes from Oscar−/− mice (Supplementary Fig. 2g, h).

We further investigated the in vivo role of OSCAR in OA caused by DMM surgery, a commonly used surgical model in mice[25] that mimics clinical meniscal injury in the development of human OA[28,29] We found that Oscar−/− mice that underwent DMM surgery exhibited significantly less articular cartilage destruction, as determined by Osteoarthritis Research Society International (OARSI) grade[25,30] and safranin-O staining compared with WT mice that underwent DMM surgery (Fig. 2a, b). Other DMM-induced OA symptoms, such as subchondral bone sclerosis, loss of hyaline cartilage, and calcification of the articular cartilage zone, were also markedly reduced in Oscar−/− mice (Fig. 2c, d). Three-dimensional microcomputed tomography (μCT) analysis showed that subchondral bone plate (SBP) thickness and bone volume were inhibited in Oscar−/− mice compared with WT mice subjected to DMM surgery (Fig. 2e). Consistent with these findings, OSCAR deficiency resulted in downregulated expression of the extracellular matrix-degrading enzymes, such as MMP3, MMP13, and ADAMTS5 (Fig. 2f, h), and upregulation of aggrecan and COL2A1 (Fig. 2g, i). These results collectively suggest that OSCAR is needed for OA pathogenesis in mice.

**OSCAR regulates TRAIL-induced chondrocyte apoptosis.** To identify regulatory factors in OA pathogenesis that are regulated by OSCAR, we generated RNA-seq data for articular cartilage obtained at 2- and 4 weeks after sham or DMM surgery in Oscar−/− and WT mice. A total of 12,597 protein-encoding genes with Transcripts Per Kilobase Million (TPM) values greater than 1 in at least one sample were used for differential gene expression analysis (Supplementary Fig. 3a, b). As candidates for OSCAR regulatory factors, we focused on the genes commonly attenuated by Oscar deficiency at 2- and 4 weeks after surgery (Supplementary Fig 3c, d). Among 1270 selected genes (Fig. 3a), functional enrichment analysis of genes whose expression was altered

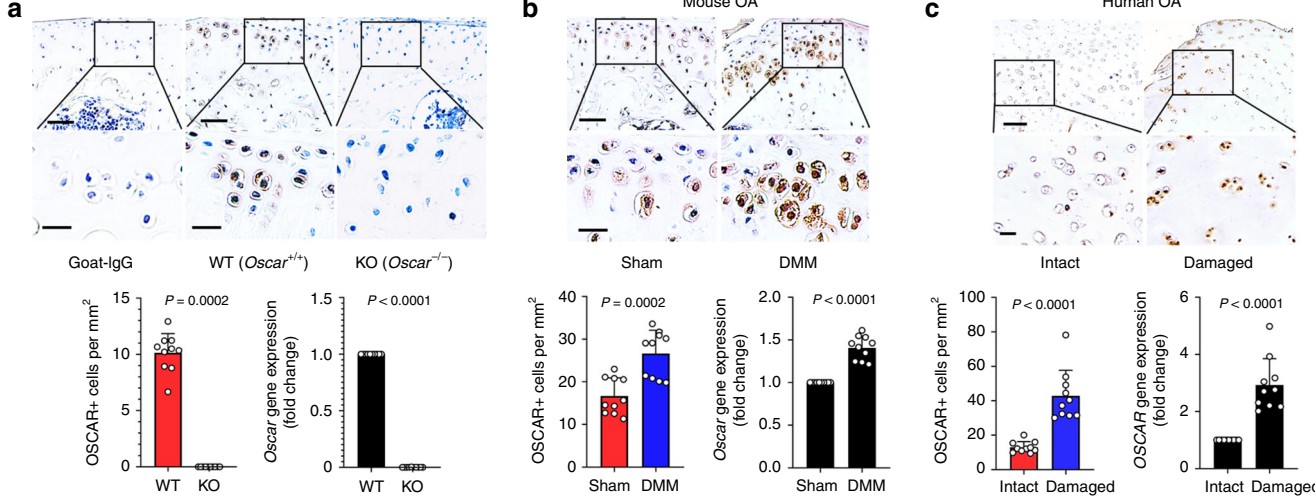

**Fig. 1 OSCAR expression in mice and humans is upregulated in articular chondrocytes in OA. a** Immunostaining for OSCAR protein (top) and mRNA levels (bottom-right) in articular chondrocytes of $Oscar^{-/-}$ and WT mice. Error bars represent mean ± S.E.M. of $n = 10$ mice. Scale bar = 50 μm (top), 20 μm (bottom). **b** IHC (top) and qRT-PCR (bottom-right) analyses of OSCAR in articular cartilage of DMM surgery mice (compared to sham-operated mice). Error bars represent mean ± S.E.M. of $n = 10$ mice. Scale bar = 50 μm (top), 20 μm (bottom). **c** Representative images show immunostaining (top) and quantitative analyses (bottom-right) of OSCAR in intact and damaged regions of human OA articular cartilage. Error bars represent mean ± S.E.M. of $n = 10$ patient samples. Scale bar = 200 μm (top), 50 μm (bottom). Student's two-tailed $t$ test was used for statistical analysis, with $p$ values displayed in figure.

by $Oscar$ deficiency revealed epithelial–mesenchymal transition (EMT), apoptosis, TGFβ signaling, IL2-STAT5 signaling, and TNFα-NFκB signaling pathways (Fig. 3b). Expression of the EMT gene, $Mmp2$ (encoding metalloprotease 2), the hypoxia gene, $Epas1$ (encoding HIF2α), and the TNFα-NFκB pathway genes, $Nos2$ and $Ptgs2$, were decreased in $Oscar^{-/-}$ mice (Supplementary Fig. 3e). These genes were already known to be associated with OA pathogenesis[12,31,32].

Network analysis showed that cell death regulatory pathways are involved in OSCAR-dependent OA (Fig. 3c). The gene encoding tumor necrosis factor-related apoptosis-inducing ligand (TRAIL; $Tnfsf10$), which functions as a cell death-inducing ligand[33,34], was one of the significantly altered genes whose expression was downregulated in OA chondrocytes in $Oscar^{-/-}$ mice (Fig. 3c). Significant downregulation of TRAIL (encoded by $Tnfsf10$) in OA chondrocytes in $Oscar^{-/-}$ mice was confirmed by quantitative reverse transcription polymerase chain reaction (Fig. 3d). DMM-induced upregulation of TRAIL levels in articular cartilage was reduced in $Oscar^{-/-}$ mice compared with that in WT mice (Fig. 3e). A decoy receptor, osteoprotegerin (OPG encoded by $Tnfrsf11b$), binds to TRAIL[35], resulting in the suppression of the function of TRAIL to induce chondrocyte apoptosis[36,37]. Thus, we investigated whether OSCAR also regulates OPG expression in OA chondrocytes. Interestingly, OPG mRNA was substantially decreased in mice with DMM-induced OA (Fig. 3f). In addition, OPG levels were also markedly decreased in the articular cartilage of mice with OA (Fig. 3g). However, we found no significantly changes in the levels of OPG mRNAs and proteins in OA chondrocytes in $Oscar^{-/-}$ mice, indicating that OSCAR deficiency in mice prevents the reduction of OPG expression (Fig. 3f, g). In addition, terminal deoxynucleotidyl transferase dUTP nick-end labeling (TUNEL) staining showed increased apoptosis in OA chondrocytes in WT mice compared with that in $Oscar^{-/-}$ mice (Fig. 3h, i). Similar to OA chondrocytes in mice, expression of $Tnfsf10$, but not $Tnfrsf11b$, as well as chondrocyte apoptosis were increased in the damaged regions of human OA patients (Supplementary Fig. 3g, h). Collectively, these results strongly suggest that OSCAR reciprocally regulates TRAIL and OPG expression in OA chondrocytes

and is thereby responsible for increased chondrocyte apoptosis during OA pathogenesis.

**OSCAR affects chondrocyte apoptosis by collagen binding**. The direct role of OSCAR in apoptosis was further examined using the primary cultures of murine articular chondrocytes. We examined OSCAR expression in primary cultures of murine chondrocytes stimulated with interleukin-1β (IL-1β) or tumor necrosis factor-α (TNF-α), both of which are OA-associated pro-inflammatory cytokines[38]. Treatment of IL-1β, but not TNF-α, moderately increased the levels of OSCAR and TRAIL (Supplementary Fig. 4a, b). However, pretreatment with collagenase, followed by IL-1β treatment, markedly increased levels of OSCAR and TRAIL, but not OPG, (Supplementary Fig. 4a, b) and further induced chondrocyte apoptosis (Fig. 4a, Supplementary Fig. 4c). Furthermore, IL-1β together with collagenase upregulated TRAIL receptor DR5 (encoded by $Tnfrsf10b$), but not DcR1 (encoded by $Tnfrsf10c$), DcR2 (encoded by $Tnfrsf10d$), or RANKL (encoded by $Tnfsf11$) (Supplementary Fig. 4d). Interestingly, a TRAIL neutralizing antibody downregulated expression of OSCAR and TRAIL as well as cell death regulatory genes in chondrocytes from WT, but not $Oscar^{-/-}$ mice, treated with IL-1β in the presence of collagenase (Supplementary Fig. 4e–g).

To examine the stimulatory effects of OSCAR on chondrocyte apoptosis, we knocked down $Oscar$ using small inhibitory RNA (siRNA). The knockdown of $Oscar$ markedly decreased mRNA and protein levels of TRAIL (Fig. 4b, c). Moreover, it inhibited IL-1β-induced chondrocyte apoptosis (Fig. 4d, e). These effects were inhibited by a soluble hOSCAR-Fc prepared by fusing the non-cytoplasmic domain of human OSCAR to the Fc part of human IgG1 (Fig. 4f, g). In addition, TUNEL staining showed that hOSCAR-Fc efficiently blocks chondrocyte apoptosis (Fig. 4h). Remarkably, we found that a minimal OSCAR-binding triple-helical peptide, followed by IL-1β treatment, upregulated TRAIL and reduced OPG. These peptide effects could be inhibited by treatment of a soluble hOSCAR-Fc (Fig. 4i, j). Collectively, these results suggest that OSCAR stimulation in chondrocytes is responsible for the coordinate increase of TRAIL relative OPG,

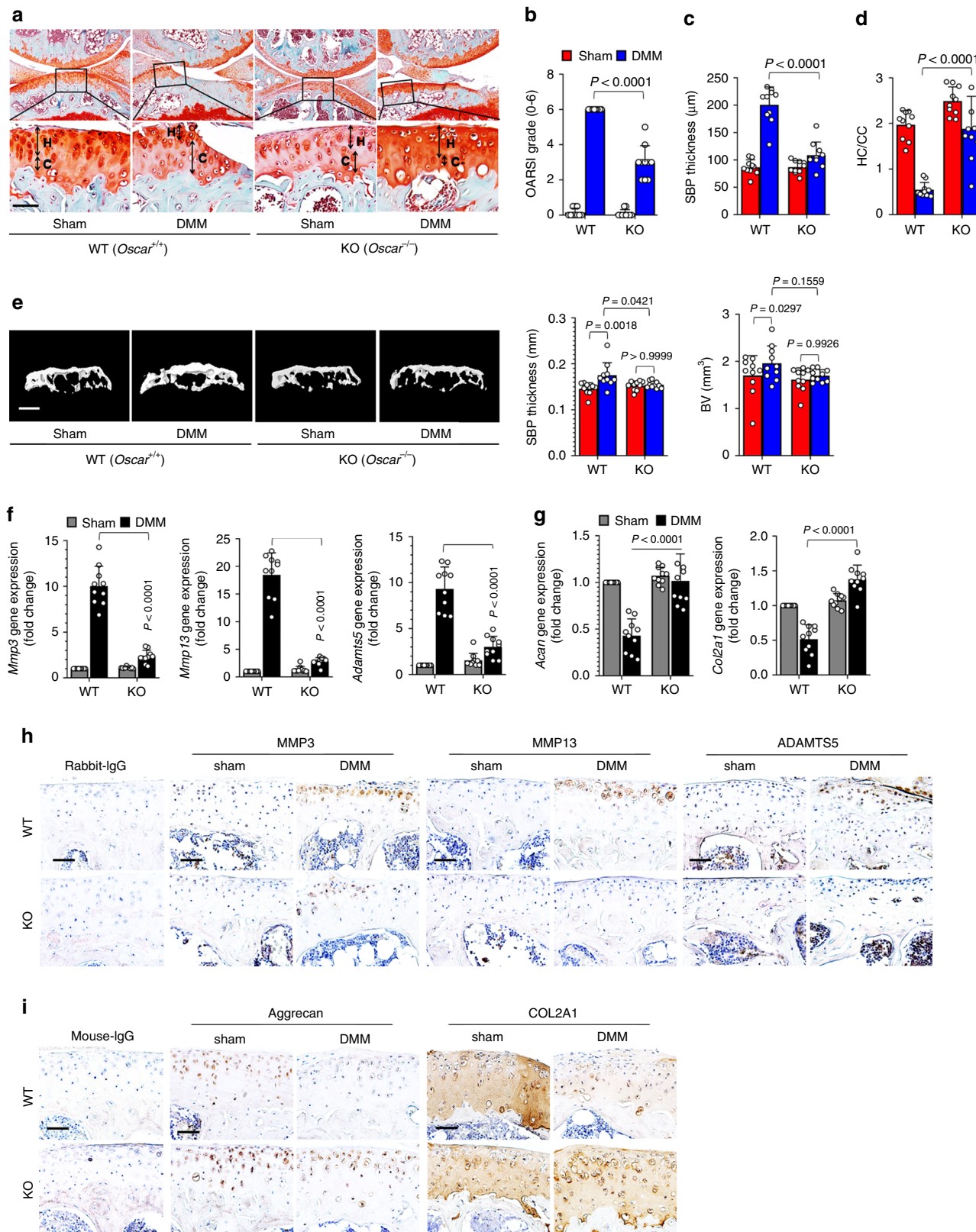

hence exerting a catabolic effect on chondrocytes under conditions of collagen degradation.

**OSCAR-Fc protects against articular cartilage degeneration**. To study the impact of OSCAR inhibition on experimental OA, we used two OA mouse models: collagenase-induced OA mice and

mice that underwent DMM surgery. We injected hOSCAR-Fc (2 mg kg$^{-1}$) or an isotype-matched IgG1 control (2 mg kg$^{-1}$) once per week for 8 weeks in DMM-operated mice. In agreement with our above data for $Oscar^{-/-}$ mice, the treatment of hOSCAR-Fc significantly reduced all collagenase- or DMM-induced manifestations of OA, including articular cartilage destruction, subchondral bone sclerosis, loss of hyaline cartilage, and calcification of the

**Fig. 2 Oscar⁻/⁻ mice exhibit reduced OA pathogenesis. a–d** Experimental OA was examined by safranin-O staining (**a**) and scoring of OA parameters, including articular cartilage destruction (OARSI grade) (**b**), SBP thickness (**c**), and ratio of hyaline cartilage (HC) to the calcified cartilage (CC) (**d**) in WT and Oscar⁻/⁻ mice. Oscar⁻/⁻ and WT mice underwent sham or DMM surgery for 8 weeks. Scale bar = 50 μm. **e** Representative 3D reconstructed micro-CT images and quantitative analysis of mouse tibia subchondral bone plates at 8 weeks post DMM surgery compared with that of the sham control. Scale bars = 1000 μm. Error bars represent mean ± S.E.M. of n = 10 mice (**b–e**). **f, g** qRT-PCR analysis of articular cartilage tissue of WT and Oscar⁻/⁻ mice subjected to sham or DMM surgery with error bars representing mean ± S.E.M. of n = 10 mice. **h, i** IHC analyses of MMP3, MMP13 and ADAMTS5 (**h**), aggrecan and COL2A1 (**i**) in OA articular cartilage from DMM surgery mice compared with sham-operated mice (n = 10). Scale bar = 50 μm. Two-way ANOVA was performed followed by Sidak's Multiple Comparison's test, with p values indicated in figure.

articular cartilage zone (Fig. 5a–d, Supplementary Fig. 5a–d). Moreover, expression of the matrix-degrading enzymes, MMP3, MMP13, and ADAMTS5 and TRAIL, were decreased in OA chondrocytes from hOSCAR-Fc-treated mice (Fig. 5e, g, Supplementary Fig. 5e), whereas expression of type II collagen, aggrecan, and OPG were increased by hOSCAR-Fc treatment (Fig. 5f, g, Supplementary Fig. 5f). Furthermore, chondrocyte apoptosis was markedly suppressed in OA chondrocytes from hOSCAR-Fc treated mice (Fig. 5h, i). The OSCAR-Fc derived from the murine Oscar sequence (mOSCAR-Fc) exerted similar inhibitory effects on experimental OA models (Supplementary Fig. 5g–i). These data suggest that the OSCAR-Fc protein exerts its disease-modifying effects on OA by blocking OSCAR.

## Discussion

In the present study, we demonstrated that the OSCAR levels increase during OA pathogenesis in human and mouse articular cartilages. Oscar deficiency in mice prevents the development of OA, lowers the expression levels of OA catabolic factors and extracellular matrix molecules. Mechanistically, we found that OSCAR regulates chondrocyte apoptosis during OA pathogenesis. Oscar knockdown or deficiency suppresses OA pathogenesis by downregulating TRAIL, and reduced expression of TRAIL in joint tissue inhibits OA cartilage destruction by blocking an apoptotic signal in chondrocytes. Furthermore, OSCAR-Fc protein inhibited OA-induced cartilage destruction in mouse models, suggesting that OSCAR may represent a therapeutic target against OA.

Collagen is the most abundant protein present in mammals[39]. In humans, various collagen subtypes are present in connective tissues of diverse organs[40] and exert an important role in cellular functions such as proliferation, migration, and apoptosis[41]. In RA, structural damage increases the turnover rate of collagen within the joint, allowing interaction with immune cells[42]. For example, monocytes expressing OSCAR have been detected in RA synovium interacting with the exposed collagen[16,22]. To date, the role of collagen-recognition receptors in OA pathogenesis has not been clearly defined. A recent report indicated that the degradation and decrease in COL2A1 may initiate and promote OA progression[43,44], which suggests that collagen degradation plays a central role in the pathogenesis of OA.

OSCAR is expressed in myeloid cells, including monocytes, macrophages, dendritic cells, and osteoclasts[18,45–47]. Osteoclasts can bind to collagens exposed on bone surfaces to promote osteoclast formation[16]. However, expression of OSCAR in chondrocytes has not been reported previously. Therefore, we hypothesized that cartilage collagen may become available for interaction with OSCAR-expressing chondrocytes and may considerably influence OA joints. Our results indicate that the collagen–OSCAR interaction resulted in chondrocyte apoptosis to promote OA development following mechanical injury induced by DMM surgery in mice. Since the DMM model is most representative of OA development following traumatic joint injury in humans[28,29], our findings suggest that traumatic OA occurs as result of collagen–OSCAR binding. It will be of interest to investigate whether OSCAR is involved in nontraumatic OA,

such as age-associated OA. Further studies on detailed OSCAR expression and function in other subsets of OA are needed to define the mechanistic basis for OA development.

To investigate the molecular mechanisms that regulate the function of OSCAR in OA, we performed RNA-seq analysis to search for OSCAR regulatory factor(s). We found that OSCAR is closely related to TRAIL-mediated chondrocyte apoptosis in OA pathogenesis. TRAIL is a type II transmembrane protein of the TNF superfamily that triggers the apoptotic signaling cascade by binding to either of the two cognate death receptors, TRAIL-R1/DR4 and TRAIL-R2/DR5, which are expressed on the cell surface[36,48,49]. Previous reports demonstrated the involvement of TRAIL in the development of collagen-induced arthritis[50,51]. In humans, normal articular chondrocytes express TRAIL receptors, thereby modulating apoptosis. Moreover, cartilage obtained from OA-induced rats showed increased expression of TRAIL[36,37]. In addition, TRAIL binds to the decoy receptors, OPG, DcR1, and DcR2, that either lack a death domain or contain a truncated death domain[35,49,52], and is thereby unable to activate an apoptotic signal. In this study, we have shown that TRAIL is downregulated in articular cartilage of Oscar-deficient mice, thus inhibiting chondrocyte cell death. Our results suggest a close regulation between OSCAR and TRAIL as well as OPG during OA pathogenesis. A recent study, using on an OPG transgenic OA mouse, suggested that the administration of OPG prevents chondrocyte apoptosis because of the ability of OPG to bind TRAIL, an inducer of chondrocyte apoptosis[53]. Interestingly, OPG was one of the altered genes whose expression was downregulated in Oscar⁻/⁻ mice in the early stage of OA (Fig. 3c). In WT mice, OPG expression was significantly decreased in the late stage of OA, but Oscar deficiency maintained OPG expression (Fig. 3f, g, Supplementary Fig. 3f). It will be important to define the temporal relationship between TRAIL and OPG in the development and progression of OA, including how OSCAR reciprocally affects their expression.

To induce apoptosis in primary chondrocytes, we treated them with IL-1β with collagenase. The effect of IL-1β on chondrocyte apoptosis is controversial and appears to depend on experimental conditions[54–56]. In humans, collagenase treatment of primary chondrocytes induces chondrocyte apoptosis, which can be inhibited by insulin-like growth factor-1[57]. In this study, we found that OSCAR is enhanced by collagenase, which causes the upregulation of TRAIL, suppression of OPG expression, and amplification of apoptotic signaling. These data suggest that OSCAR may contribute to OA by epitope exposure of collagen. Degradation of collagen fibers by collagenase generally leads to exposure of embedded sequences in cartilage which can cause an interaction with OSCAR and consequently apoptosis of chondrocytes. We confirmed that the minimal binding motif of collagen induces OSCAR expression and higher TRAIL-mediated chondrocyte apoptosis. In conclusion, collagen–OSCAR binding can be considered as a damage-related molecular pattern signal in chondrocytes to induce OA pathogenesis.

There are several potential limitations to our study. First, it was previously shown that the development of histologic OA following DMM in mice is associated with poor functional outcomes

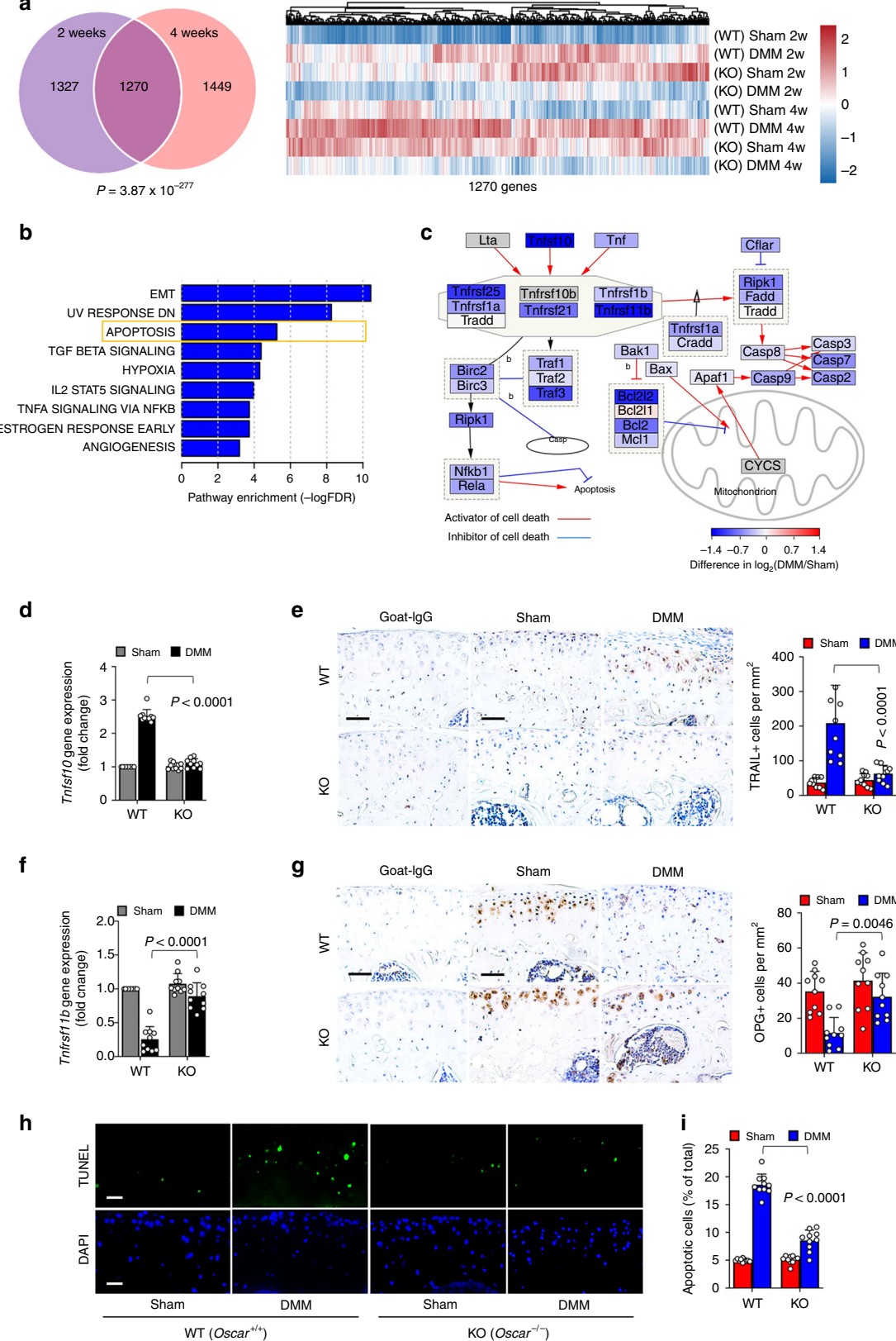

including pain[58,59]. The data presented in this study demonstrate that the collagen–OSCAR interaction contributes to cartilage and joint degeneration following DMM, and consistent with prior works[58,59]. Such pathologic changes are expected to result in pain and impaired motile function. Future studies are required to further determine the role of OSCAR in OA-associated pain and joint dysfunction. Second, while our findings demonstrate that OSCAR is increased in OA chondrocytes in humans and in murine chondrocytes stimulated with the OA-associated pro-inflammatory cytokine IL-1β, further investigation is needed to identify the transcription factor(s) responsible for IL-1β-dependent OSCAR upregulation.

**Fig. 3 OSCAR regulates OA pathogenesis via TRAIL-induced articular chondrocyte apoptosis. a** Selection of putative *Oscar* targets involved in OA pathogenesis. Venn diagram indicating the number of genes attenuated in articular cartilage *Oscar*$^{-/-}$ mice after DMM surgery (left). Hypergeometric *p* value measuring the significance of enrichment is depicted. Gene-wise scaled gene expression patterns of 1270 common genes in each sample are shown (right). **b** Functional annotations significantly enriched for 1270 common genes. Hypergeometric tests were performed using the hallmark gene annotation in MsigDB, yielding enrichment scores, defined as $-\log_{10}$ (FDR). **c** Simplified apoptotic signaling pathway altered by *Oscar* deficiency. Each gene is colored with the DIF score, defined as the extent to which gene expression is altered by *Oscar* deficiency. A negative DIF indicates transcriptional suppression in *Oscar*$^{-/-}$ mice. The network was visualized by Cytoscape v3.7 software. **d, e** qRT-PCR (**d**) and IHC (**e**) analyses of TRAIL in OA articular cartilage from DMM surgery mice compared to sham-operated mice. Scale bar = 50 μm. **f, g** mRNA levels (**f**) and immunostaining (**g**) for OPG in articular cartilage tissue from sham surgery or DMM surgery mice. Scale bar = 50 μm. **h, i** Apoptotic articular chondrocytes were detected and quantified by TUNEL assay. Scale bar = 25 μm. Error bars represent mean ± S.E.M. of *n* = 10 mice (**d–g**, **i**). Two-way ANOVA was performed followed by Sidak's Multiple Comparison's test, with *p* values indicated in figure.

In addition, we developed OSCAR-Fc fusion proteins that inhibit pathological OA pathogenesis. Although OA, the most common form of arthritis, is the leading cause of disability, there is no effective disease-modifying therapy. Thus, the development of strategies to disrupt the OSCAR–collagen interaction with biologics, such as Fc fusion proteins and neutralizing antibodies, or small molecules may provide potential therapeutics for OA.

Collectively, our findings demonstrate that dysregulation of the OSCAR–collagen interaction caused by OSCAR deficiency affects chondrocyte apoptosis via the downregulation of TRAIL expression in OA pathogenesis, and OSCAR-Fc fusion proteins blocked OA pathogenesis. Thus, our results suggest that OSCAR acts as a catabolic regulator of cartilage degeneration via chondrocyte apoptosis, and support that OSCAR could be a therapeutic target for OA.

## Methods

**Experimental OA in mice**. C57BL/6J (WT) male mice and OSCAR-deficient mice (*Oscar*$^{-/-}$, C57BL/6) were used for experimental OA studies. We purchased 8–10-week-old C57BL/6 male mice from Japan SLC, Inc. (Hamamatsu, Japan); C57BL/6 *Oscar*$^{-/-}$ mice have been described previously[16]. Experimental OA was induced by IA injection of collagenase[27,28] in 8–10-week-old male mice or by DMM surgery[25,28,29] in 10–12-week-old male mice. To induce collagenase-induced OA, mice were injected with 10 U of collagenase dissolved in 10 μl phosphate-buffered saline (pH 7.4) at the right knee joint space through the IA space along the patellar tendon. The collagenase type VII (from *Clostridium histolyticum*, enzyme activity > 500 U) was purchased from Sigma-Aldrich (St Louis, MO, USA). In DMM surgery, the medical meniscus ligament was surgically removed from the right knee joint of hind limb. Mice injected with phophate0buffered saline or sham-operated mice were used as controls. The mice were euthanized at 1, 2, 4, or 8 weeks after OA induction. To study the protective effects of OSCAR-Fc from induced OA, we prepared soluble hOSCAR-Fc fusion protein by fusing the non-cytoplasmic domain of human OSCAR to the Fc part of human immunoglobulin G1 (IgG1). We injected 2 mg kg$^{-1}$ of hOSCAR-Fc fusion protein or 2 mg kg$^{-1}$ of an isotype-matched IgG1 control once per week for 4 or 8 weeks, beginning 1 week after IA collagenase injection or DMM surgery, respectively. The mice were euthanized at 5 or 9 weeks after OA induction and subjected to biochemical and histological analyses. All mice were maintained in pathogen-free barrier facilities. Mice were housed in barrier facility at 5 or less per cage at 24–26 °C with humidity ranging between 30 and 60% with 12 h light/dark cycles. For each experiment, age- and sex-matched mice were used and randomly allocated to each experimental group. All animal experiments were approved by the Institutional Animal Care and Use Committees (IACUC) (Protocol No: IACUC 19-027) of Ewha Womans University and followed National Research Council Guidelines.

**Human samples**. Cartilage tissues with OARSI grade 6 were obtained from ten patients with OA ranging in age from 63 to 78 years (three males and seven females) during total knee replacement surgery. To rule out the effects of other underlying diseases, OA patients had no RA, metabolic disease, or other inflammatory diseases at the time of surgery (Source data Fig. 1c). The institutional review board of Yonsei University (Protocol No: IRB 3-2018-0251), Gangnam Severance Hospital, South Korea, approved the use of the articular cartilage. All participants provided informed consent.

**Micro-CT analyses**. To analyze knee joints, samples were isolated from 10- to 12-week-old male WT and *Oscar*$^{-/-}$ mice subjected to DMM surgery. The samples were fixed in 10% formaldehyde and scanned using a Skyscan 1176 in vivo μCT scanner (Bruker micro-CT, Kontich, Belgium) at 75 kV and 333 μA. A total of 360°

views were acquired at 0.7° angle increments, each for an exposure time of 260 ms, to give a resolution of 18 μm. The raw data from the tibiae acquired by micro-CT were translated into two-dimensional cross-sectional gray scale image slices using NRecon (Bruker microCT, Kontich, Belgium). From the acquired two-dimensional images, structural parameters of the tibial trabecular bone were evaluated by a CT Analyzer (CT-AN, v1.10.9.0, Bruker microCT, Kontich, Belgium), including SBP thickness (μm) and the bone volume fraction (BV, mm$^3$).

**Histology and immunohistochemistry**. At the end of the experiments, the knee joints of the mice were fixed in 10% formaldehyde at 4 °C for >48 h, decalcified in 0.5 M ethylenediaminetetraacetic acid in PBS (pH 7.4) for 14 days, and embedded in paraffin. Next, the paraffin blocks were cut into 5 μm sections and stained with hematoxylin and eosin and safranin-O. Articular cartilage destruction was scored using OARSI grades (0–6), a standard OA-grading system[25,30], while sclerosis and articular cartilage destruction were identified by safranin-O staining and measured using OsteoMeasureXP (OsteoMetrics, Inc., Atlanta, GA, USA), Adobe photoshop (v19.1.3, San Jose, CA, UA), and an Olympus DP72 charge-coupled device camera (v2.1, Olympus Corporation, Tokyo, Japan). Subchondral bone sclerosis was determined by measuring the SBP thickness[3]. Immunostaining was performed using a standard protocol. The knee joint sections were incubated with primary anti-OSCAR antibody (Cat# SC34235, Santa Cruz Biotechnology, 1:200 dilution) at 4 °C overnight. To detect immunoactivity, we used a 3,3′-diaminobenzidine peroxidase (horseradish peroxidase) substrate detection kit (Vector Laboratories, Inc., Burlingame, CA, USA) and counterstained with hematoxylin. Additional immunostaining was performed using antibodies against MMP3 (Cat# Ab53015, Abcam, 1:50 dilution), MMP13 (Cat# Ab51072, Abcam, 1:25 dilution), aggrecan (Cat# Ab1031, Abcam, 1:100 dilution), COL2A1 (Cat# MAB8887, Sigma-Aldrich, 1:50 dilution), ADAMTS5 (Cat# GTX100332, Genetex, 1:50 dilution), OPG (Cat# sc8468, Santa Cruz Biotechnology, 1:200 dilution), and TRAIL (Cat# AF1121, R&D System, 1:200 dilution). Articular chondrocyte apoptosis was determined using the TUNEL assay and a kit from Millipore (Temecula, CA, USA). The specimens were visualized under a fluorescence microscope and the number of apoptotic articular chondrocytes in relation to the total number of cells was counted.

**Primary culture of articular chondrocytes**. Articular chondrocytes were isolated from femoral condyles and tibial plateaus of 4–5-day-old Institute of Cancer Research mice by digestion with 0.2% collagenase type II[60]. The chondrocytes were maintained in Dulbecco's modified Eagle's medium (DMEM; HyClone, Logan, UT, USA) containing 10% fetal bovine serum and were treated with 5 ng mL$^{-1}$ of IL-1β or 50 ng mL$^{-1}$ of TNF-α for 48 h. *Oscar*-specific siRNAs were designed with the coding sequences of mouse *Oscar*, which are shown in Supplementary Table 1. Chondrocytes were cultured for 60 h and transfected for 7–8 h with siRNAs using Lipofectamine 2000 (Invitrogen, Carlsbad, CA, USA). Transfected cells were exposed for 2 h with 50 U mL$^{-1}$ collagenase type VII (Sigma-Aldrich, St Louis, MO, USA), before being treated with IL-1β for 48 h. Non-silencing siRNA was used as a negative control. Articular chondrocytes were isolated from femoral condyles and tibial plateaus of 2–3-day-old WT and *Oscar*$^{-/-}$ mice. Chondrocytes were exposed for 2 h with 50 U mL$^{-1}$ collagenase, then preincubated for 30 min with 1 μg mL$^{-1}$ TRAIL neutralizing antibody (Cat# 14-5951-82, Thermo Fisher Scientific), before being treated with IL-1β. Rat IgG2a kappa isotype (Cat# 12-4321-80, Thermo Fisher Scientific) was used as a negative control. Chondrocytes were incubated for 2 h with 5 μM of an OSCAR-binding triple-helical peptide, before being treated with IL-1β or 10 μg mL$^{-1}$ hOSCAR-Fc fusion protein. Articular chondrocyte apoptosis was determined using the TUNEL assay and a kit from Millipore (Temecula, CA, USA). The OSCAR-binding triple-helical peptide was synthesized by Peptron (Daejeon, Korea), and the purity was analyzed using high-performance liquid chromatography. A triple-helical-forming peptide[16,19,20] composed of the minimal binding motif (GPO)$_3$GPOGPAGFO(GPO)$_2$G was used for western blot and quantitative reverse transcription polymerase chain reaction analyses.

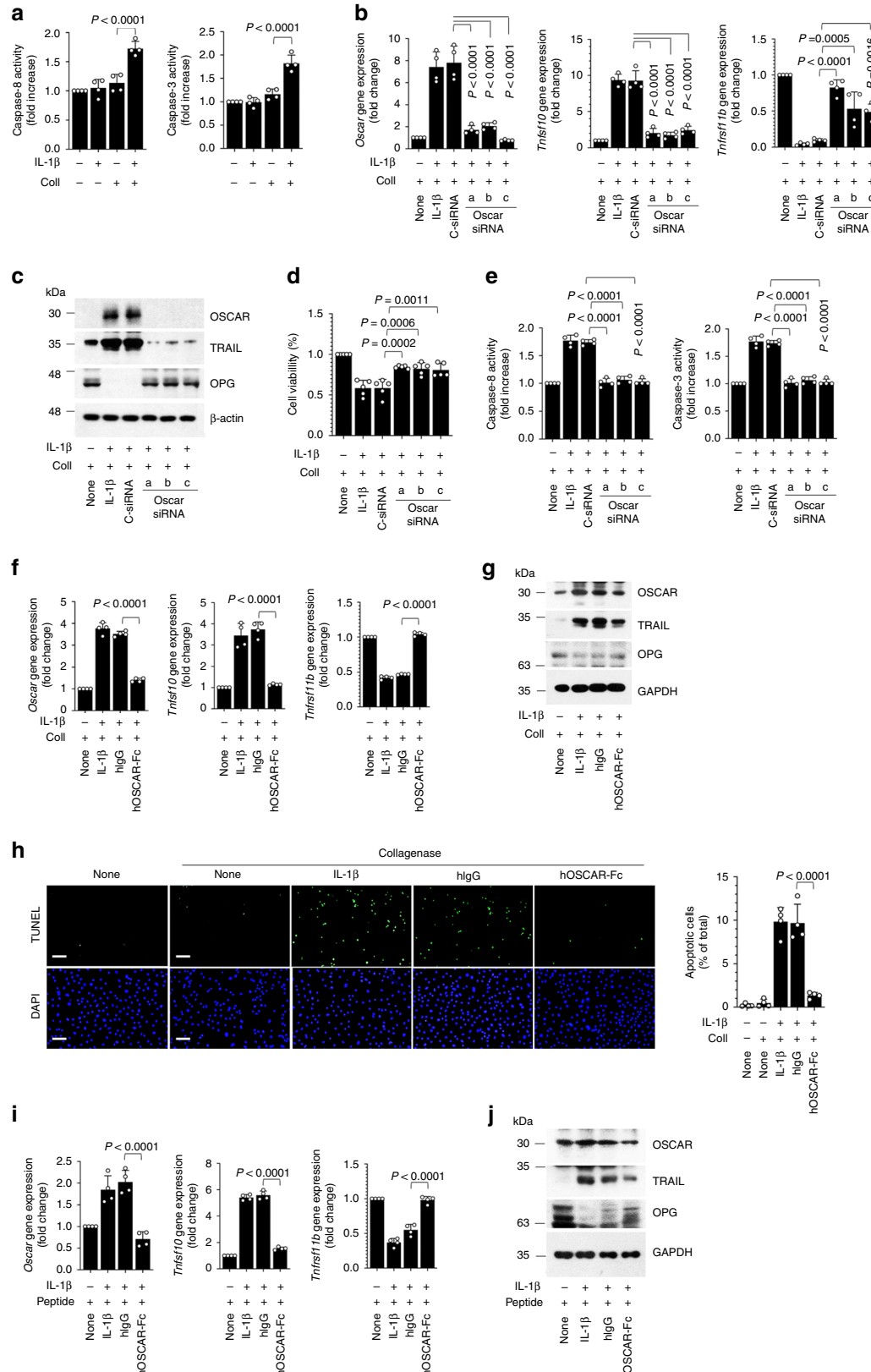

**Generation of hOSCAR-Fc fusion protein**. For these experiments, 293F cells were transfected with a gene encoding the dimeric form of the extracellular domains (amino acids 19–233) of human OSCAR with the Fc region of human IgG1 cloned into pVITRO1-Fc. The secreted purified protein (hOSCAR-Fc) was loaded into a Thermo Scientific™ Pierce™ Protein G-Sepharose bead column (Thermo Fisher Scientific, Waltham, MA, USA) and eluted with an elution buffer (100 mM glycine, pH 2.0, and 1 M Tris-Cl, pH 7.0; Duchefa Biochemie B.V., Haarlem, the Netherlands) to immediately neutralize the protein. The tubes containing high concentrations of protein were collected, extensively dialyzed against PBS, and kept frozen at −80 °C.

**RNA isolation and real-time PCR**. Total RNAs were isolated from primary chondrocytes using TRIzol reagent (Invitrogen, Carlsbad, CA, USA) or from

**Fig. 4 OSCAR–collagen binding co-stimulates IL-1β-induced apoptotic signaling in articular chondrocytes. a** Caspase activity in mouse articular chondrocytes treated with IL-1β (5 ng mL$^{-1}$) and collagenase (Coll, 50 U mL$^{-1}$). **b, c** qRT-PCR analysis (**b**) and western blotting (**c**) in mouse articular chondrocytes. Chondrocytes were either untreated or transfected with control siRNA (C-siRNA) (100 nM) or siRNAs specific for mouse *Oscar* and exposed to IL-1β (5 ng mL$^{-1}$) and collagenase (50 U mL$^{-1}$) for 48 h. **d** Articular chondrocyte viability was quantified by MTT assay. Error bars represent mean ± S.E.M. of $n = 5$ wells. **e** Caspase-8 and caspase-3 activity was measured using the respective assay kits. **f, g** qRT-PCR analysis (**f**) and western blotting (**g**) in mouse articular chondrocytes. Chondrocytes were treated with hIgG or hOSCAR-Fc (10 µg ml$^{-1}$) with collagenase and exposed to IL-1β for 48 h. **h** Apoptotic articular chondrocytes were detected and quantified by TUNEL assay. Scale bar = 100 µm. **i, j** qRT-PCR analysis (**i**) and western blotting (**j**) in mouse articular chondrocytes treated with 10 µM OSCAR-binding triple-helical peptide before IL-1β treatment. Articular chondrocytes were treated with hOSCAR-Fc for 48 h. Error bars shown in **a**, **b**, **e**, **f**, **h** and **i** are mean ± S.E.M. for $n = 4$ independent experiments. One-way ANOVA was performed followed by Dunnett's Multiple Comparison's test (**a**, **i**) and Tukey's Multiple Comparison's test (**b**, **d**, **e**, **f**, **h**), with $p$ values indicated in figure.

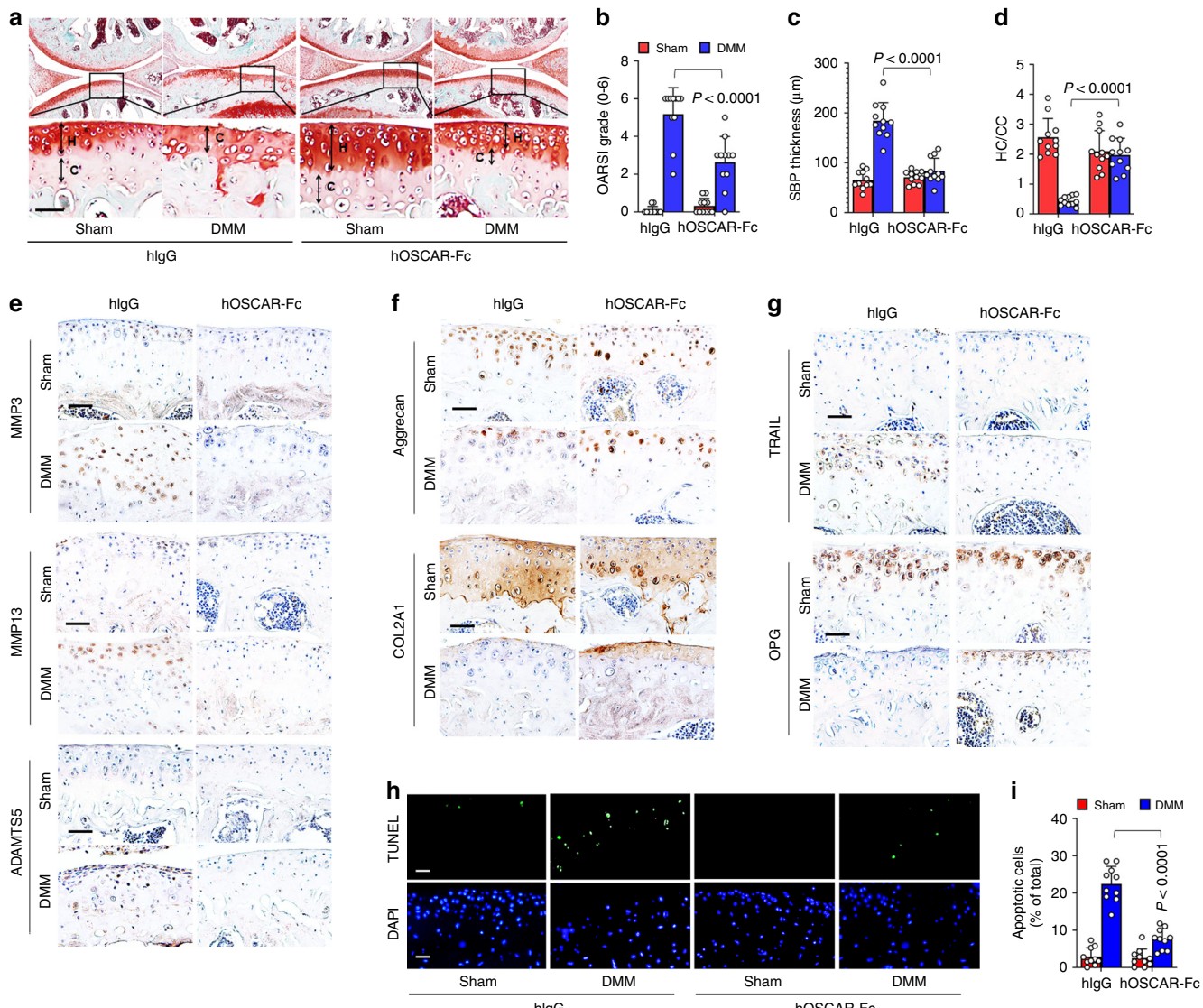

**Fig. 5 OSCAR-Fc blocks OA pathogenesis in mice.** WT mice subjected to sham or DMM surgery were IA-injected with human IgG as a control or hOSCAR-Fc (2 mg kg$^{-1}$) to block OSCAR in joint tissue. Articular cartilage sections were subjected to safranin-O staining (**a**) and quantitative analysis of OARSI grade (**b**), SBP thickness (**c**), and ratio of hyaline cartilage (HC) to calcified cartilage (CC) (**d**). Scale bar = 50 µm. Error bars represent mean ± S.E.M. of $n = 10$ mice (**b–d**). **e–g** IHC analyses of MMP3, MMP13, and ADAMTS5 (**e**), aggrecan and COL2A1 (**f**), and TRAIL and OPG (**g**) in articular cartilage tissue from sham surgery or DMM surgery mice ($n = 10$). Scale bar = 50 µm. **h, i** Apoptotic articular chondrocytes were detected and quantified by TUNEL assay. Scale bar = 25 µm. Error bars represent mean ± S.E.M. of $n = 10$ mice (**i**). Two-way ANOVA was performed followed by Sidak's Multiple Comparison's test, with $p$ values indicated in figure.

mouse and human knee joint tissues using the RNA Mini Kit (Life Technologies, Carlsbad, CA, USA) and reverse-transcribed to generate complementary DNA (cDNA) using the Superscript cDNA synthesis kit (Invitrogen) according to the manufacturer's instructions. Real-time PCR was performed using the KAPA SYBR Green fast qPCR kit (Kapa Biosystems, Inc., Wilmington, MA, USA) on a StepOnePlus real-time PCR machine (Applied Biosystems, Foster City, CA, USA). The samples were analyzed in triplicate and data were normalized to β-actin mRNA expression. See Supplementary Table 2 for primer sequences.

**Western blotting analysis and immunoprecipitation**. Articular chondrocytes were extracted with lysis buffer (50 mM Tris-HCl, pH 8.0, 150 mM NaCl, 0.5% deoxycholate acid, and 1% NP-40) containing protease and phosphatase inhibitors. For western blotting analysis, antibodies raised against OSCAR (Cat# PA5-47171, Thermo Fisher Scientific, 1:1000 dilution), OPG (Cat# sc8468, Santa Cruz Biotechnology, 1:1000 dilution and Cat# ab183910, Abcam, 1:1000 dilution), and TRAIL (Cat# ab10516, Abcam, 1:1000 dilution) were used. In addition, β-actin (Cat# sc47778, Santa Cruz Biotechnology, 1:1000 dilution) and glyceraldehyde-3-phosphate dehydrogenase (Cat# sc32233, Santa Cruz Biotechnology, 1:1000 dilution) antibody was used as a loading control. Source data file contains the uncropped gel scans of western blots.

**Caspase assay**. Caspase-3 and caspase-8 activities were determined using a caspase colorimetric assay kit (Biovision Research Products, Milpitas, CA, USA)[60]. Briefly, articular chondrocyte lysates were incubated with a substrate for caspase-3 (DEVD-pNA) or caspase-8 (IETD-pNA) at 37 °C for 2 h. The activity of each caspase was determined by measuring the absorbance at 405 nm.

**MTT assay**. Primary articular chondrocytes were transfected with control short interfering RNA, or si-$Oscar$ in the presence of 50 U mL$^{-1}$ of collagenase and 5 ng mL$^{-1}$ of IL-1β for 48 h. Then, 3-(4,5-dimethylthiazol-2-yl)-2,5-diphenyltetrazolium bromide (MTT) was added to the culture medium in each well, and the samples were incubated at 37 °C for 4 h. Dimethyl sulfoxide was added to each well at room temperature for 10 min, after which absorbance was measured at an optical density of 570 nm.

**RNA sequencing and data analysis**. Total RNA was extracted using the RNA Mini Kit (Life Technologies, Carlsbad, CA, USA) from knee joint tissues of WT and $Oscar^{-/-}$ mice euthanized at 2 or 4 weeks after sham or DMM surgery. Each group consisted of a combination of ten murine articular cartilages. Sequencing libraries were prepared according to the manufacturer's instructions (TruSeq Stranded mRNA Library Prep Kit; Illumina, San Diego, CA, USA). Paired-end sequencing of 101-mer read length was performed using a HISEQ 2500 sequencing system (Illumina). The sequencing quality of raw FASTQ files was assessed using FastQC (https://www.bioinformatics.babraham.ac.uk/projects/fastqc/). Low-quality reads and adapter sequences in reads were eliminated using BBDuk (http://jgi.doe.gov/data-and-tools/bb-tools/). Trimmed reads were aligned to the GRCm38 genome reference using STAR aligner (v2.6.0a). Gene level TPM was calculated using RSEM (v1.2.17) with Gencode v22 annotation. FASTQ files and processed data are available from the Gene Expression Omnibus (GEO: GSE147529) database.

To identify $Oscar$ downstream mechanisms involved in OA pathogenesis, we roughly selected genes that met the following four requirements: (i) $\log_2(E_{DMM}/E_{sham})$ in $Oscar^{-/-}$ mice < 0, where $E_x$ was the gene expression value (TPM) in cartilage undergoing x-operations; (ii) $\log_2(E_{DMM}/E_{sham})$ in WT mice > 0; (iii) DIF < DIF$_{25\%}$, where DIF was defined as the difference between $\log_2(E_{DMM}/E_{sham})$ in KO mice and $\log_2(E_{DMM}/E_{sham})$ in WT mice, and DIF$_{25\%}$ was the lower quartile value of DIF values for a total of 12,597 genes; and (iv) all three requirements (i)–(iii) were met in the data obtained at 2 and 4 weeks. For functional enrichment analysis for the 1270 genes, we performed hypergeometric tests using the hallmark gene set in the Molecular Signatures Database (http://software.broadinstitute.org/gsea/msigdb). Subsequent multiple testing corrections were done through false discovery rate estimation. Annotated human genes were mapped to mouse genes using MGI (http://www.informatics.jax.org) human/mouse orthology information. The network was visualized by Cytoscape v3.7 software[61].

**Quantitation and statistical analysis**. All data were collected from at least four independent experiments. Statistical analysis was performed using a Student's two-tailed $t$ test to analyze differences between two samples. The analysis of more than two samples was performed using one-way and two-way analysis of variance (ANOVA), followed by pairwise multiple comparisons if significant. Data based on the comparison of two samples with a variable measured using an ordinal grading system were analyzed using the Sidak's multiple comparisons test. Multiple comparisons were performed using the Dunnett's test or Tukey's test and one-way ANOVA. The sample size for each experiment was not predetermined. $P$ values are indicated in the figures or in source data, and the error bars represent standard error of the mean (S.E.M.) for parametric data and calculated 95% confidence intervals for non-parametric data. All the graphs and statistical analysis were performed using GraphPad Prism (v8.1.2, San Diego, CA, USA).

**Reporting summary**. Further information on research design is available in the Nature Research Reporting Summary linked to this article.

## Data availability
The data that support the findings of this study are available within the article and its Supplementary Information files or from the corresponding author upon reasonable request. RNA seq of articular cartilage in WT and $Oscar^{-/-}$ mice can be obtained from the Gene Expression Omnibus (GEO) database with accession number GSE147529. The following figures have associated raw data: Figs. 1a–c, 2b–h, 3d–i, 4a, b, d–f, h, i, and

5b–f; Supplementary Figs. 1a–g, 2b–h, 3f, g, i, 4b–d, f, and 5b–i. Source data are provided with this paper.

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

## Acknowledgements

This work was supported by grants from the National Research Foundation of Korea funded by the Korea Government (MSIP) (no. 2017M3A9C8030537; no. 2017R1A6A3A11031928; no. 2019R1A5A6099645; and no. 2019R1A6C1010020). We would like to thank Yongwon Choi (University of Pennsylvania, USA) for helpful discussions and suggestions. We thank Jang-Soo Chun (GIST, South Korea) for discussions and help in in vitro studies.

## Author contributions

D.R.P. performed the experiments, analyzed the data, and co-wrote the paper. J.K., G.M.K., and M.K. performed in vitro and in vivo experiments. G.M.K. and H.S. generated human and mouse OSCAR-Fc fusion proteins. H.L. and W.K. analyzed the RNA-seq data. D.H., H.L., and H.S.K. performed and analyzed μCT data. M.C.P. evaluated human samples. S.Y.L. designed the study, analyzed, and interpreted the data, and co-wrote the paper.

## Competing interests

The authors declare no competing interests.
