## [Peer Review File · Nature Communications]

Reviewers' Comments:

Reviewer #1:

Remarks to the Author:

In this manuscript Lee et al. report that osteoclast-associated receptor blockade prevents articular cartilage destruction via chondrocyte apoptosis regulation. Authors suggest that OSCAR deletion abrogates OA manifestations. Besides, treatments with human OSCAR-Fc fusion protein attenuated OA pathogenesis caused by experimental OA. However, it is not exactly clear what is the rationale to expect a role for OSCAR in development of OA? Moreover, the power of some of the analyses seems rather small which could lead to unreliable results.

Major comments

1. In Fig. 1b and c, the immunohistochemical images of whole layers of the cartilage samples need to be presented. And at what stage of OA does OSCAR expression manifest? Do you need ECM degradation before expression?
2. The authors used the complete knock out of OSCAR, what are the phenotypes of other organs? Do they affect the phenotype of arthritis? What is the phenotype of animal at different stages of embryonic development and post embryonic stages? This is precisely to know the time frame where OSCAR is crucial for the cartilage development.
3. In Fig. 2, given that OSCAR plays a role in subchondral bone sclerosis, the micro-CT data should be added.
4. OA patients feel pain during their physical actions like running, walking, etc. OSCAR knockout mice have any such impaired motile actions?
5. In Fig. 2e-f and Supplementary Fig. 1e-f, the author found that MMP-3, MMP-13, ADAMTS5, Col2a1 and aggrecan expression were regulated by qPCR, it may be more convincing to confirm by IHC.
6. In Fig. 3a, gene expression changes of seven selected genes known to initiate the apoptotic signaling cascades were shown, however, the seven altered genes whose expression were not all downregulated in OA chondrocytes in Oscar^{-/-} mice, such as Igf2, Igf1, FasI, Tnf and Lta. Only Tnfsf 10 and Ngf are downregulated, suggesting the apoptotic signaling pathway was not the main pathway of Oscar inducing OA. In Fig. Supplementary Fig. 2a, significantly enriched pathways also included hypoxia and others, the altered genes of each pathway screened need to be validated by qPCR. Then the main pathway that OSCAR regulates OA pathogenesis would be found.
7. In Fig. 3a and Supplementary Fig. 2b, the author indicated n=10 per group, but they just presented one lane of each group heatmap, please show the whole lanes, this will be helpful for knowing the changes of gene expressions.
8. In Fig. 4, with respect to mechanism, the research relatively simple, which transcription factor plays a role in IL-1b inducing OSCAR expression? And the specific location of its promoter? Activities of OSCAR in OA chondrocytes? These need to be elucidated.
9. In Fig. 5 and Supplementary Fig. 3, OSCAR-Fc blocks OA pathogenesis in mice. the expression of MMP-3, MMP-13, ADAMTS5, Col2a1 and aggrecan and genes of signaling pathways screened need to be shown. Also the physical actions of mice need to be shown.

Minor comments

1. MW markers to the immunoblots.
2. The discussion could have been more logical as their mechanistic details are not very appealing.

Reviewer #2:

Remarks to the Author:

The manuscript submitted by Park et al. studied the pathogenesis of osteoarthritis (OA) and identified OSCAR as potential therapeutic option. OSCAR would be increased in OA models and in human samples whereas TRAIL would be increased. The depletion of OSCAR (OSACR KO mice model) abrogated OA clinical manifestations and could be explained by a decrease of TRAIL

dependent chondrocyte apoptosis. OA mice treated with recombinant human OSCAR attenuated OA.

In the past decades, even numerous scientific papers described the main dysregulatory cellular and molecular networks in OA, no efficient therapeutic option has been proposed and new therapeutic options are still needed. In this context, the present paper is interesting. However, the mode of action of OSCAR in OA should be better studied and most of results presented were obtained from mouse samples and there is no clear demonstration of OSCAR in human OA. Furthermore, their relationship between OSCAR and TRAIL should be better analysed.

Figure 1b: Authors showed that OSCAR is expressed by chondrocytes. Lower magnification should be presented and a limited area of the articular cartilage. Did the authors find any expression gradient in the articular cartilage?

Figure 1c: there is no clear information about the patients enrolled in the study. More specifically, there is high probability that the mean age of patients enrolled in OA group (damaged) was significantly higher than healthy donors (Intact). That is a crucial information because Crotti et al (Arthritis Res Ther 2012, 14(6): R245) analyzed OSCAR expression in OA, in inactive/active rheumatoid arthritis (RA) and healthy cartilage and found a higher expression in active RA than inactive RA and OA which showed similar expression level. In normal tissues (« Healthy tissue ») they did not find any expression of OSCAR however the mean age of the donors was half compared to the pathological groups. In addition, the work previously published has already demonstrated that OSCAR was increased in OA compared to normal group. Based on this observation, OSCAR expression should be analysed in comparable groups (CT and OA) and the number of patient samples analyzed which is currently very low and must be increased. The correlation between OSCAR and patient ages must be taken in consideration before to reach any definitive conclusion.

Figure 2/Supplementary Fig 1: due to the discrepancy already observed in the literature between transcript and protein expression for collagens, analysis of the protein levels should be studied.

Figure 3: there is partial description of RNAsequencing data. The number of modulated genes (up and down) is not clearly described. There is partial information in the figure legends. Please show a Volcano plot and the heatmap integrating all modulated genes must be shown. The selection of 7 genes as shown in Figure 3a is not extremely objective and the clusterisation of these genes is not so convincing on the base of the data shown after 2 and 4 weeks. Did the authors identify any cluster genes from all genes modulated between WT and KO cartilage? The results of RNAseq of OA in CT versus OSCAR KO mice must be shown. In addition, heatmap analysis must be presented in Sham group. Indeed, authors indicated that RNAseq data from WT and OSCAR^{-/-} mice subjected to sham and DMM surgery were presented, but it seems that only OA after 2 or 4 weeks were shown. This point must be clarified. It is mandatory to validate the data obtained from mouse samples in human samples (CT and OA).

Figure 3a: RNAseq do not show any modulation of OPG which do not appear in the list of the 7 genes identified.

Please show lower magnification for all IHC carried out.

Fig 3b-e: at which time point has been carried the experiments shown (2 or 4 weeks)

Figures 3 and 4: authors speculated on the functional relationship between OSCAR and TRAIL. They analysed OSCAR, TRAIL and OPG expression after collagenase treatment but did not show the same investigation in the absence of collagenase treatment.

Technical remarks: siRNA can induce off target effects and conventionally it is strongly recommended to confirm the effects with a minimum of 3 different sequences of siRNA.

OPG possesses a heparin binding domain leading to its potential binding to GAG, collagens, etc. Il-

1 has been shown to upregulated OPG expression that is not found in Figure 4b. Collagenase treatment could explain this discrepancy however OPG transcript should be studied as well RANKL and LGR4. The study of TRAIL function must integrate all protagonists of the TNF/TNR superfamily members related to its regulation. TRAIL silencing in chondrocytes should be assessed as well as the comparison between WT and OSCAR KO mice.

What was the impact of the collagenase treatment on chondrocyte adherence and viability by using microscopic approaches?

The investigations of cell apoptosis limited to caspase-3 must be extended to molecular networks conventionally observed in apoptosis.

Figure 5: authors showed that OSCAR can block OA in mice. OSCAR should counterbalance its deficiency in OSCAR KO mice. This experiment should be carried out.

Why to use 2 mg/Kg of recombinant OSCAR. How this dose has been defined?

Reviewer #3:

Remarks to the Author:

The submitted manuscript investigates the role of osteoclast-associated receptor (OSCAR) in osteoarthritis (OA). The manuscript demonstrate that OSCAR is more abundance in chondrocyte of the damaged cartilage of OA in human and mice models, that ablation of OSCAR using knockouts or hOSCAR-Fc prevent the manifestation of OA induced by surgical destabilization of the medial meniscus (DMM) and intra-articular injection of collagenase in vivo, and that OSCAR co-stimulate IL-1 β -induced apoptotic signaling in aricular chondrocytes. The manuscript was overall a pleasure to read, and is in my eyes novel, well presented and nearly complete.

1. In the abstract, the authors state: "We identified an unexpected function of OSCAR...". Why unexpected, bearing in mind the recent studies in RA and osteoclasts? I believe it's very much in line with the recent studies, providing a novel extension to OA.

2. In figure 1, the OSCAR immunoreactivity and gene expression is addressed in cartilage from human OA patients. Here the authors subdivide the cartilage in intact and damaged cartilage. Please clarify the definition of these regions according to the OARSI grade, as this must be variable between the five patients? Or even better scale the immunoreactivity according to different degrees of damage according to the OARSI grade, not just two extremes? Low magnification images of the whole cartilage and the regions investigate would help. Are the damaged regions also showing more apoptotic chondrocytes? How was the regions divided for the gene expression analysis? Note that OSCAR in the y-axis of the left panel C graph should be capital, as it's in human cartilage this is investigated.

3. In figure 4, it's clear that western blot are cropped, and that several band are present outside the illustrated field. Please provide the whole western blots as supplementary data, and edit the cropping in figure 4. Gadph can be cropped much more, giving more space to Oscar, Trail and Opg. Please also present western blot with positive and negative controls in the supplementary figure (see also next point).

4. Validation of antibodies used for immunostaining and western blots. The authors only refer on the statement from the manufacture, which is too often fare from the truth. The reliability of the antibodies needs to be validated either with other antibodies binding other regions of OSCAR, TRAIL and OPG or by in situ hybridization. The inclusion of negative control tissues from knockout mice would provide a clear validation of the antibodies specificity.

5. In figure 5, the mice experiments only include five mice per group. This is too few, especially when bearing in mind the variability observed in the OARSI score of the DMM groups. I'm not convinced that the control versus hOSCAR-Fc is different if one don't assume that they are normally distributed, which I don't believe to be the case. Please include additional mice in each group.

6. The color codes of the graphs are very nice, but not completely consistent. Please use the gray/black bars consistently for all gene expression analysis and blue/red bars for all histology including the TUNEL analysis.

In response to Reviewer #1's comments:

In this manuscript Lee et al. report that osteoclast-associated receptor blockade prevents articular cartilage destruction via chondrocyte apoptosis regulation. Authors suggest that OSCAR deletion abrogates OA manifestations. Besides, treatments with human OSCAR-Fc fusion protein attenuated OA pathogenesis caused by experimental OA. However, it is not exactly clear what is the rationale to expect a role for OSCAR in development of OA? Moreover, the power of some of the analyses seems rather small which could lead to unreliable results.

Major comments

Q1. *In Fig. 1b and c, the immunohistochemical images of whole layers of the cartilage samples need to be presented. And at what stage of OA does OSCAR expression manifest? Do you need ECM degradation before expression?*

Reply: We appreciate your critical review and believe that addressing these comments will improve the quality of our manuscript.

In response to your comments, we have changed magnification to low for all immunohistochemical (IHC) images performed to represent whole layers of articular cartilage. In addition, we increased the number of human OA samples analyzed from n = 5 to n = 10. Accordingly, the previous IHC images in Fig. 1a-c were replaced with new images. Quantitative graphs showing OSCAR+ cells and *Oscar* expression levels were also changed to new graphs in Fig. 1c.

We also performed additional experiments to determine the OA stage at which OSCAR was expressed. As shown in the IHC images in Supplementary Fig. 1b, it is difficult to observe OSCAR-expressing cells one week after DMM surgery. However, OSCAR+ cells began to appear 2 weeks after surgery, and after 4 weeks, most cells expressed OSCAR. We observed that OSCAR mRNA expression, as shown in Fig. 1c, was similar to that determined by IHC. On the other hand, we found that the representative cartilage-degrading enzymes, Mmp3 and Mmp13, were expressed 4 weeks after DMM surgery (Supplementary Fig. 1d-e). Taken together, OSCAR may play a role in the early stages of OA development. This point has been described in the revised manuscript (Page 5, lines 14-18) as follows:

Furthermore, OSCAR was expressed in the early stages of OA, and it was further induced in later stages (Supplementary Fig. 1a, b). However, the expression of matrix-degrading enzymes, including matrix metalloproteinases-3 and -13 (MMP3 and MMP13), increased at 4 weeks following surgery (Supplementary Fig. 1d-g). These findings suggest that OA-associated catabolic events in mice and humans upregulate OSCAR expression in the early stages of OA.

Q2. The authors used the complete knock out of OSCAR, what are the phenotypes of other organs? Do they affect the phenotype of arthritis? What is the phenotype of animal at different stages of embryonic development and post embryonic stages? This is precisely to know the time frame where OSCAR is crucial for the cartilage development.

Reply: We appreciate the reviewers' comments. After the initial discovery of OSCAR as a costimulatory receptor in osteoclast precursors (Kim N et al., J Exp Med 195:201, 2002), it has been suggested that OSCAR's role in human disease may be related to osteoporosis (Kim et al., J Bone Miner Res 20:1342, 2005) and rheumatoid arthritis (Herman et al., Arthritis Rheum 58: 3041, 2008; Crotti et al., Arthritis Res Ther 14:R245, 2012). However, Barrow et al. reported that *Oscar* knock-out (KO) mice did not show any difference in skeletal phenotypes when compared to wild-type (WT) mice (J Clin Invest 121:3505, 2011). They suggested that OSCAR may contribute to osteoclast formation in disease conditions, such as rheumatoid arthritis (Herman et al., Arthritis Rheum. 58:3041, 2008). Nevertheless, no phenotype of OSCAR KO mice in other organs has been reported to date. There are also no reports regarding pathogenesis in OSCAR KO mice.

In our experience, breeding between *Oscar* heterozygous mutant mice, even homozygous mutant mice, gave rise to viable homozygous KO off-springs which grew normally, were fertile, and did not show any obvious phenotype. Expected numbers of mutant mice were consistently obtained in Mendelian crosses between heterozygous *Oscar* mice. Additionally, H&E staining showed no obvious abnormalities in the brain, thymus, heart, lung, liver, spleen, kidney, and testis of *Oscar* KO mice compared with those of WT mice (Author response Fig. 1). Based on previous reports and our observations, we believe that OSCAR function is dispensable for normal mouse development, growth, fertility, and skeletal development.

Author response Fig. 1. H&E staining of the multiples tissues from *Oscar* KO and WT mice at 8-weeks old. Scale bar = 200 μ m.

Since OA is an age-associated disease, we observed a spontaneous OA development with age between OSCAR KO and WT mice as determined by OARSI grade and OSCAR expression (Author response Fig. 2). There was no difference in OA development in 8-week-old mice. However, we found that OSCAR KO mice at 6 months of age exhibited less articular cartilage destruction compared with WT mice exhibiting increased OSCAR expression and OA development.

In this manuscript, we have used two OA disease models: collagenase-induced OA (CIOA) and destabilization of the medial meniscus (DMM). While no mouse model

perfectly mimics OA pathogenesis in humans, the DMM model is highly representative of a significant subset of human OA and is also the gold standard in the field for experimental investigation of OA (Lorenz J. and Grassel S., *Methods Mol Biol.* 1194:401, 2014; Culley et al., *Methods Mol Biol.* 1226:143, 2015). In addition, the DMM model is most representative of OA development following post-traumatic joint injury (PTOA) in humans.

With regard to the timing of OSCAR expression during OA development, OSCAR is likely to play a role in both traumatic and age-associated OA. Further studies on detailed OSCAR expression and function in other subsets of OA are needed to define the mechanistic basis for OA development.

Author response Fig. 2. OSCAR expression is upregulated in articular chondrocytes in age-associated OA. (a) Age-associated OA was examined by safranin-O staining and scoring of OA parameters, OARSI grade, in WT and *Oscar*^{-/-} mice (*n* = 6). Scale bar = 200 μ m. **(b)** Immunohistochemical analyses of OSCAR in articular cartilage in WT and *Oscar*^{-/-} mice (*n* = 6). Scale bar = 50 μ m. Values are expressed as mean \pm SEM with two-way ANOVA. *n* indicates the number of biologically independent samples.

We address this issue in the following text added to the Discussion (Page 12, lines 3-11).

Our results indicate that the collagen-OSCAR interaction resulted in chondrocyte apoptosis to promote OA development following mechanical injury induced by DMM surgery in mice. Since the DMM model is most representative of OA development following traumatic joint injury in humans^{28,29}, our findings suggest that traumatic OA occurs as result of collagen-OSCAR binding. It will be of interest to investigate whether OSCAR is involved in non-traumatic OA, such as age-associated OA. Further studies on detailed OSCAR expression and function in other

subsets of OA are needed to define the mechanistic basis for OA development.

Q3. *In Fig.2, given that OSCAR plays a role in subchondral bone sclerosis, the micro-CT data should be added.*

Reply: We appreciate the reviewer's comment and believe that this analysis is meaningful to observe OSCAR's role in subchondral bone sclerosis. The data are now shown in Fig. 2e and described in the main text (page 6, lines 21-23) as follows:

Three-dimensional microcomputed tomography (μ CT) analysis showed that subchondral bone plate thickness and bone volume were inhibited in *Oscar*^{-/-} mice compared with WT mice subjected to DMM surgery (Fig. 2e).

Q4. *OA patients feel pain during their physical actions like running, walking, etc. OSCAR knockout mice have any such impaired motile actions?*

Reply: We appreciate the reviewer's suggestion. The primary objective of this study was to identify the new role of OSCAR in OA pathogenesis.

As previously demonstrated in both Huesa et al. (Ann Rheu Dis 75:1989, 2016) and McCulloch et al. (Osteoarthritis Cartilage 27:1800, 2019), the cartilage degeneration and joint breakdown that occurs following DMM results in increased pain. They assessed OA-related pain by measuring weight bearing before and after DMM surgery. Based on the previous reports, we believe that histologic demonstration of cartilage degeneration, subchondral bone sclerosis, and hyaline cartilage loss is the most definitive outcome in the DMM model, and as a result, this was the focus of our studies.

In this manuscript, we provide extensive evidence based on histologic results that OSCAR plays critical roles in the cartilage breakdown and joint tissue destruction that are characteristic of OA. Given that previous reports demonstrated that histologic OA pathology is associated with pain and dysfunction in mice (Huesa et al., 2016; McCulloch et al., 2019), we do not believe that the assessment of pain (by measuring weight bearing) would significantly advance the conclusions of our study. We now address this potential limitation of our study in the Discussion (Page 14, lines 3-10) as follows:

There are several potential limitations to our study. First, it was previously shown that the development of histologic OA following DMM in mice is associated with poor functional outcomes including pain^{59,60}. The data presented in this study demonstrate that the collagen-OSCAR interaction contributes to cartilage and joint degeneration following DMM, and consistent with prior works^{59,60}. Such pathologic changes are expected to result in pain and impaired motile Function. Future studies are required to further determine the role of OSCAR in OA-associated pain and joint dysfunction.

Q5. In Fig.2e-f and Supplementary Fig. 1e-f, the author found that MMP-3, MMP-13, ADAMTS5, Col2a1 and aggrecan expression were regulated by qPCR, it may be more convincing to confirm by IHC.

Reply: According to the reviewer's comments, we have confirmed the expression of matrix-degrading enzymes by IHC. The data now added in revised manuscript in Fig. 2h, i and Supplementary Fig. 2g, h.

Q6-1. In Fig.3a, gene expression changes of seven selected genes known to initiate the apoptotic signaling cascades were shown, however, the seven altered genes whose expression were not all downregulated in OA chondrocytes in *Oscar*^{-/-} mice, such as *Igf2*, *Igf1*, *Fasl*, *Tnf* and *Lta*. Only *Tnfsf10* and *Ngf* are downregulated, suggesting the apoptotic signaling pathway was not the main pathway of *Oscar* inducing OA.

Q6-2. In Fig. Supplementary Fig. 2a, significantly enriched pathways also included hypoxia and others, the altered genes of each pathway screened need to be validated by qPCR. Then the main pathway that OSCAR regulates OA pathogenesis would be found.

Reply: We appreciate the reviewers' suggestions for these two related issues. To address these, we have analyzed the RNA sequencing data (new Figures 3a-c and Supplementary Figures 3a-g). Network analyses as shown in Fig. 3c suggested that cell death regulatory pathways are involved in OSCAR-induced OA. Our data further suggest that TRAIL, functioning as a cell death ligand, was one of the significantly altered genes whose expression was downregulated in OA chondrocytes in *Oscar* KO mice (Fig. 3c).

In response to the reviewer's comment Q6-2, we validated several of the altered genes from each pathway by qPCR analysis in articular cartilage from WT and *Oscar* KO mice subjected DMM surgery. As shown in Supplementary Fig. 3e, *Mmp2* (encoding MMP2), *Epas1* (encoding HIF2 α), *Nos2* (encoding NO synthase), and *Ptgs2* (encoding COX-2) were decreased in *Oscar* KO mice.

In addition, we further validated the expression of cell death regulatory genes, including *Tnfsf10* (encoding TRAIL), *Tnfrsf11b* (encoding OPG), *Casp3*, *Casp8*, *Bax*, and *Bcl2* in human OA and control samples (Supplementary Fig. 3g).

Taken together, we describe these points more clearly in the revised text as follows (Page 7, lines 5-24; Page 8, lines 1-2):

To identify regulatory factors in OA pathogenesis that are regulated by OSCAR, we generated RNA-seq data for articular cartilage obtained at 2- and 4-weeks after sham or DMM surgery in *Oscar*^{-/-} and WT mice. A total of 12,597 protein-encoding genes with Transcripts Per Kilobase Million (TPM) values greater than 1 in at least one sample were used for differential gene expression analysis (Supplementary Fig. 3a, b). As candidates for OSCAR regulatory factors, we focused on the genes

commonly attenuated by *Oscar* deficiency at 2- and 4-weeks after surgery (Supplementary Fig 3c, d). Among 1,270 selected genes (Fig. 3a), functional enrichment analysis of genes whose expression was altered by *Oscar* deficiency revealed epithelial-mesenchymal transition (EMT), apoptosis, TGF β signaling, IL2-STAT5 signaling, and TNF α -NF κ B signaling pathways (Fig. 3b). Expression of the EMT gene, *Mmp2* (encoding metalloprotease 2), the hypoxia gene, *Epas1* (encoding HIF2 α and the TNF α -NF κ B pathway genes, *Nos2* and *Ptgs2*, were decreased in *Oscar*^{-/-} mice (Supplementary Fig. 3e). These genes were already known to be associated with OA pathogenesis^{12,32,33}.

Network analysis showed that cell death regulatory pathways are involved in OSCAR-dependent OA (Fig. 3c). The gene encoding tumor necrosis factor-related apoptosis-inducing ligand (TRAIL; *Tnfsf10*), which functions as a cell death- inducing ligand^{34,35}, was one of the significantly altered genes whose expression was downregulated in OA chondrocytes in *Oscar*^{-/-} mice (Fig. 3c).

Q7. In Fig. 3a and Supplementary Fig. 2b, the author indicated n=10 per group, but they just presented one lane of each group heatmap, please show the whole lanes, this will be helpful for knowing the changes of gene expressions.

Reply: For RNA seq analysis, we originally compared gene expression profiles between 2-, 4-, and 7-weeks after DMM surgery. However, seven weeks after DMM surgery, we realized that a sufficient amount of RNA suitable for RNA seq analysis could not be obtained due to severe cartilage degradation. Therefore, we combined articular cartilages from ten mice for each group at 2- and 4-weeks after DMM surgery and analyzed eight different groups as shown in Fig. 3a and Supplementary Fig. 3a.

As previously addressed in Q6-1 and Q6-2, a total of 12,597 protein encoding genes were used for differential gene expression analysis from 2- and 4- weeks after sham or DMM surgery in *Oscar* KO and WT mice (Supplementary Fig. 3a). As candidates for OSCAR targets, we focused on those genes commonly attenuated by *Oscar* deficiency at 2- and 4- weeks after surgery. We also presented gene-wise scaled gene expression patterns of 1,270 common genes in each sample (Fig 3a).

Q8. In Fig.4, with respect to mechanism, the research relatively simple, which transcription factor plays a role in IL-1 β inducing OSCAR expression? And the specific location of its promoter? Activities of OSCAR in OA chondrocytes? These need to be elucidated.

Reply: We appreciate the reviewer's critical comments. Previous reports have indicated that IL-1 β increases HIF-2 α expression in murine chondrocytes (Yang et al., Nat Med 16:687, 2010; Saito et al., Nat Med 16:678, 2010) and HIF-2 α directly induces the expression of catabolic factors, including MMP1, MMP3, MMP9, MMP12, MMP13,

ADAMTS4, NOS2, and PTGS2 in OA chondrocytes. Thus, HIF-2 α appears to act as a master transcription factor during OA development.

Our data also indicated that OSCAR deficiency resulted in downregulation of HIF-2 α target genes such as *Mmps*, *Nos2*, and *Ptgs2* as well as even *Hif2 α* itself. Thus, we hypothesized that IL-1 β -induced HIF-2 α may be responsible for OSCAR induction by an auto-amplification loop in chondrocytes.

To test this, we overexpressed HIF-2 α in primary chondrocytes by using adenovirus. As shown in Author Response 3, HIF-2 α overexpression induced MMP3, but not OSCAR, suggesting that IL-1 β -dependent HIF-2 α is not involved in OSCAR expression.

Author response Fig. 3. HIF-2 α does not regulates OSCAR expression. qRT-PCR analysis in mouse articular chondrocytes. Chondrocytes were either untreated or infected with 800 multiplicity of infection (MOI) of control adenovirus or Ad-*Epas1* ($n = 4$). Values are expressed as the mean \pm SEM with two-way ANOVA. n indicates the number of biologically independent samples.

We agree with the reviewer that the elucidation of OSCAR's activities in chondrocytes should be addressed. In this revised manuscript, we have performed two approaches by using OSCAR siRNAs and human OSCAR-Fc protein to examine OSCAR activity primarily in chondrocytes.

We have tested the effects of three different OSCAR siRNAs on gene expression regulation and chondrocyte viability in Fig. 4b-e and found that OSCAR knockdown regulates IL-1 β -induced apoptotic signaling in chondrocytes. Similar to OSCAR knockdown, OSCAR-Fc treatment downregulates apoptotic signaling, thereby inhibiting chondrocyte cell death (Fig. 4f-j).

It is probable that the regulation of chondrocyte apoptosis by OSCAR, as identified in this study, may be part of OSCAR's function or activity at the cellular and organism level. As previously answered in Q2, according to our knowledge, what is known about OSCAR function is very limited, and its role in chondrocytes is completely unknown. Thus, we believe that further study on OSCAR at the cellular and pathophysiological level is needed in the future.

Q9. In Fig.5 and Supplementary Fig. 3, OSCAR-Fc blocks OA pathogenesis in mice. the expression of MMP-3, MMP-13, ADAMTS5, Col2a1 and aggrecan and genes of signaling pathways screened need to be shown. Also the physical actions of mice need to be shown.

Reply: In response to the reviewer's comments, we have carried out additional experiments to examine the expression of MMP3, MMP-3, ADAMTS5, aggrecan, and COL2A1 by IHC and added these data in Fig. 5e, f and Supplementary Fig. 5e, f. We also analyzed the expression of TRAIL and OPG by IHC and presented the data in Fig. 5g.

As previously answered in *Q4*, given that previous reports demonstrated that histologic OA pathology is associated with pain and dysfunction in mice (Huesa et al., 2016; McCulloch et al., 2019), we do not believe the assessment of physical actions (by measuring weight bearing) would significantly advance the conclusions of our study. We address this potential limitation of our study in the Discussion (Page 14, lines 3-10) as follows:

There are several potential limitations to our study. First, it was previously shown that the development of histologic OA following DMM in mice is associated with poor functional outcomes including pain^{59,60}. The data presented in this study demonstrate that the collagen-OSCAR interaction contributes to cartilage and joint degeneration following DMM, and consistent with prior works^{59,60}. Such pathologic changes are expected to result in pain and impaired motile Function. Future studies are required to further determine the role of OSCAR in OA-associated pain and joint dysfunction.

Minor comments

Q1) MW markers to the immunoblots.

Reply: We have corrected this and added whole blots to the Supplementary data.

Q2) The discussion could have been more logical as their mechanistic details are not very appealing.

Reply: We appreciate the reviewer's comment. However, the findings from our new experiments further strengthen our conclusions that OSCAR promotes OA pathogenesis and OSCAR could be a new therapeutic option for OA. This is the first report on the role of OSCAR in OA pathogenesis. Nevertheless, the revised manuscript needs for further investigation of the mechanistic role of OSCAR in regulating OA pathogenesis.

We have discussed the strengths and weaknesses of this study as logically as possible in

this revised manuscript.

In response to Reviewer #2's comments:

The manuscript submitted by Park et al. studied the pathogenesis of osteoarthritis (OA) and identified OSCAR as potential therapeutic option. OSCAR would be increased in OA models and in human samples whereas TRAIL would be increased. The depletion of OSCAR (OSACR KO mice model) abrogated OA clinical manifestations and could be explained by a decrease of TRAIL dependent chondrocyte apoptosis. OA mice treated with recombinant human OSCAR attenuated OA.

In the past decades, even numerous scientific papers described the main dysregulatory cellular and molecular networks in OA, no efficient therapeutic option has been proposed and new therapeutic options are still needed. In this context, the present paper is interesting. However, the mode of action of OSCAR in OA should be better studied and most of results presented were obtained from mouse samples and there is no clear demonstration of OSCAR in human OA. Furthermore, there relationship between OSCAR and TRAIL should better analysed.

Q1. *Figure 1b: Authors showed that OSCAR is expressed by chondrocytes. Lower magnification should be presented and a limited area of the articular cartilage. Did the authors find any expression gradient in the articular cartilage?*

Reply: We appreciate your critical review and believe that addressing these comments will improve the quality of our manuscript.

In response to your comments, we have now presented a low magnification for all immunohistochemical (IHC) images carried out to represent whole layers of the articular cartilage. Accordingly, the previous IHC images Fig. 1a-c were replaced with new ones.

In regard to the expression gradient of OSCAR, we examined OSCAR expression by IHC at each time point after DMM surgery. We found that OSCAR expression began to appear 2-weeks after surgery in the upper layer of the hyaline cartilage, and after 4-weeks, most cells of the hyaline cartilage express OSCAR. Because hyaline cartilage decreased as OA progressed, OSCAR was eventually expressed in the entire cartilage, including calcified cartilage. The new data were added to Supplementary Fig. 1a, b.

Q2. *Figure 1c: there is no clear information about the patients enrolled in the study. More specifically, there is high probability that the mean age of patients enrolled in OA group (damaged) was significantly higher than healthy donors (Intact). That is a crucial information because Crotti et al (Arthritis Res Ther 2012, 14(6): R245) analyzed OSCAR expression in OA, in inactive/active rheumatoid arthritis (RA) and healthy*

cartilage and found a higher expression in active RA than inactive RA and OA which showed similar expression level. In normal tissues (« Healthy tissue ») they did not find any expression of OSCAR however the mean age of the donors was half compared to the pathological groups. In addition, the work previously published has already demonstrated that OSCAR was increased in OA compared to normal group. Based on this observation, OSCAR expression should be analyzed in comparable groups (CT and OA) and the number of patient samples analyzed which is currently very low and must be increased. The correlation between OSCAR and patient ages must be taken in consideration before to reach any definitive conclusion.

Reply: We appreciate the reviewer's critical comments. First, we have analyzed 5 additional patient samples and for a total of 10 patients. The ages of the patients in the group ranged from 63 to 78 years with an OARSI grade of 6. Undamaged (Intact control) and damaged samples from each patient were obtained during total knee replacement surgery.

Second, to rule out the effects of other underlying diseases, OA patients had no rheumatoid arthritis, metabolic disease, or other inflammatory diseases at the time of surgery. We included detailed information on the enrolled patients in Source data Fig. 1c.

Accordingly, the previous IHC images in Fig. 1c were replaced with new ones. Quantitative graphs representing OSCAR-positive cells and *Oscar* expression levels were also changed to new graphs in Fig. 1c.

This point has been described in the revised Methods (Page 17, lines 1-5) as follows:

Cartilage tissues with OARSI grade 6 were obtained from 10 patients with OA ranging in age from 63 to 78 years (3 males and 7 females) during total knee replacement surgery. To rule out the effects of other underlying diseases, OA patients had no rheumatoid arthritis, metabolic disease, or other inflammatory diseases at the time of surgery (Source data Fig. 1c).

Q3. *Figure 2/Supplementary Fig 1: due to the discrepancy already observed in the literature between transcript and protein expression for collagens, analysis of the protein levels should be studied.*

Reply: In response to the reviewer's comment, we performed new IHC staining for MMP3, MMP13, ADAMTS5, aggrecan, and COL2A1. IHC analysis revealed decreased expression of MMP3, MMP13, and ADAMTS5, but upregulation of aggrecan and COL2A1 in OA chondrocytes from OSCAR KO mice. We now provide new data demonstrating IHC of the OA markers in the WT and OSCAR KO mice from DMM surgery in Fig. 2h, i and Supplementary Fig. 2g, h. We added the following text to the Results (Page 6, lines 9-12) as follows:

Similarly, immunohistochemical (IHC) analysis revealed decreased expression of MMP3, MMP13 and ADAMTS5, and upregulation of

aggrecan and COL2A1 in OA chondrocytes from *Oscar*^{-/-} mice (Fig. 2h, i, Supplementary Fig. 2g, h).

Q4. *Figure 3: there is partial description of RNA sequencing data. The number of modulated genes (up and down) is not clearly described. There is partial information in the figure legends. Please show a Volcano plot and the heatmap integrating all modulated genes must be shown. The selection of 7 genes as shown in Figure 3a is not extremely objective and the clusterisation of these genes is not so convincing on the base of the data shown after 2 and 4 weeks. Did the authors identify any cluster genes from all genes modulated between WT and KO cartilage? The results of RNAseq of OA in CT versus OSCAR KO mice must be shown. In addition, heatmap analysis must be presented in Sham group. Indeed, authors indicated that RNAseq data from WT and OSCAR^{-/-} mice subjected to sham and DMM surgery were presented, but it seems that only OA after 2 or 4 weeks were shown. This point must be clarified. It is mandatory to validate the data obtained from mouse samples in human samples (CT and OA).*

Reply: We deeply appreciate and agree with the reviewer's comments that we presented partial description of RNA sequencing data in the previous manuscript. In response to the reviewer's comments, we have now removed the previous Fig. 3a and revised these points as follows:

1. We added a full description of the RNA seq analysis in the Methods section (Page 22, lines 4-24, Page 23, lines 1-11).
2. We produced RNA seq data of articular cartilages obtained at 2- and 4-weeks after sham or DMM surgery in *Oscar* KO and WT mice. We now present gene expression values for a total of 12,597 protein-coding genes with a heatmap plot (Supplementary Fig. 3a). Of these, we selected 1,270 genes commonly attenuated at 2- and 4- weeks after DMM surgery in *Oscar* KO mice and also presented this genes expression data as a heatmap plot (Fig. 3a). We also demonstrate gene expression changes in Supplementary Fig. 3b with MA plots (Maza et al., *Commun Integr Biol* 6:e25849, 2013; McDermaid et al., *Briefings in Bioinformatics* 20:2044, 2019) which is a very useful tool for comparing differential gene expression in RNA seq experiments. A volcano plot is used to identify changes in RNA seq data sets composed of replicate data. However, for the datasets without replicate values, it is difficult to estimate the statistical significance of gene expression changes. Alternatively, we visualized the data with an MA plot (x-axis: average expression, y-axis: fold-change) to identify genes exhibiting consistent fold-changes in the 2- and 4-week data sets (Supplementary Figure 3b).
3. We performed functional enrichment analysis of genes whose expression was altered by *Oscar* deficiency. The analysis showed that epithelial-mesenchymal transition (EMT), apoptosis, TGFβ signaling, IL2-STAT5 signaling, and TNFα-NFκB signaling pathways are involved (Fig. 3b). We confirmed that the expression of the EMT gene, *Mmp2*, the hypoxia gene, *Epas1*, and the TNFα-

NFκB pathway genes, *Nos2* and *Ptgs2*, were decreased in *Oscar* KO mice (Supplementary Fig. 3e).

4. We further performed network analysis to examine whether cell death regulatory pathways are involved in OSCAR-dependent OA (Fig. 3c). We identified that TRAIL, functioning as a ligand that induces cell death, was one of the significantly altered genes whose expression was downregulated in OA chondrocytes in *Oscar* KO mice (Fig. 3c).

Based on our new analysis of RNA seq data above, we validated gene expression profiles of cell death regulatory molecules in OA chondrocytes from WT and *Oscar* KO mice (Fig. 3d-g, Supplementary Fig. 3e-f) as well as from human OA samples (Supplementary Fig. 3g). We believe that OSCAR regulates chondrocytes apoptosis through TRAIL expression regulation. All together, we described these points more clearly in the revised text as follows (Page 7, lines 7-24; Page 8, lines 1-2):

To identify regulatory factors in OA pathogenesis that are regulated by OSCAR, we generated RNA-seq data for articular cartilage obtained at 2- and 4- weeks after sham or DMM surgery in *Oscar*^{-/-} and WT mice. A total of 12,597 protein-encoding genes with Transcripts Per Kilobase Million (TPM) values greater than 1 in at least one sample were used for differential gene expression analysis (Supplementary Fig. 3a, b). As candidates for OSCAR regulatory factors, we focused on the genes commonly attenuated by *Oscar* deficiency at 2- and 4-weeks after surgery (Supplementary Fig 3c, d). Among 1,270 selected genes (Fig. 3a), functional enrichment analysis of genes whose expression was altered by *Oscar* deficiency revealed epithelial-mesenchymal transition (EMT), apoptosis, TGFβ signaling, IL2-STAT5 signaling, and TNFα-NFκB signaling pathways (Fig. 3b). Expression of the EMT gene, *Mmp2* (encoding metalloprotease 2), the hypoxia gene, *Epas1* (encoding HIF2α and the TNFα-NFκB pathway genes, *Nos2* and *Ptgs2*, were decreased in *Oscar*^{-/-} mice (Supplementary Fig. 3e). These genes were already known to be associated with OA pathogenesis^{12,32,33}.

Network analysis showed that cell death regulatory pathways are involved in OSCAR-dependent OA (Fig. 3c). The gene encoding tumor necrosis factor-related apoptosis-inducing ligand (TRAIL; *Tnfsf10*), which functions as a cell death- inducing ligand^{34,35}, was one of the significantly altered genes whose expression was downregulated in OA chondrocytes in *Oscar*^{-/-} mice (Fig. 3c).

Regarding qPCR analysis of cell death regulatory genes, such as *Tnfsf10*, *Tnfrsf11b*, *Cas3*, *Cap8*, *Bax*, and *Bcl2* in human OA samples (Supplementary Fig. 3g), the result has been described in the main text (Page 8, lines 17-19) as follows:

Similar to OA chondrocytes in mice, expression of *Tnfsf10*, but not *Tnfrsf11b*, as well as chondrocyte apoptosis were increased in the damaged regions of human OA patients (Supplementary Fig. 3g, h).

Q5. Figure 3a: RNAseq do not show any modulation of OPG which do not appear in the list of the 7 genes identified.

Reply: We appreciate the reviewer's critical comment in regard to OPG expression in RNA seq data.

First, as we responded in Q4, we have removed the previous Fig. 3a because of its ambiguity in the presented gene list.

Second, we added *Tnfrsf11b* (gene encoding OPG) as a node for the network visualizing apoptotic signaling cascade in Fig 3c. We noted that *Tnfrsf11b* was down-regulated by OSCAR deficiency. It should be noted that RNA seq data were obtained at 2- and 4-weeks after DMM surgery. However, mRNA levels and IHCs of OPG in Fig. 3f and 3g were analyzed samples from 8-weeks after DMM surgery. Therefore, these discrepancies might be due to the expression kinetics of OPG. To clarify this, we performed additional experiments to see TRAIL and OPG expression at 1-, 2-, 4- and 8-weeks after DMM surgery. As shown in supplementary Fig. 3f, we found different expression profiles between TRAIL and OPG. Importantly, OPG was regulated very differently between *Oscar* KO and WT mice 8-weeks after DMM surgery. We now address this possibility in the following newly added text in the Discussion (Page 13, lines 6-12):

Interestingly, OPG was one of the altered genes whose expression was downregulated in *Oscar*^{-/-} mice in the early stage of OA (Fig. 3c). In WT mice, OPG expression was significantly decreased in the late stage of OA, but *Oscar* deficiency maintained OPG expression (Fig. 3f, g, Supplementary Fig. 3f). It will be important to define the temporal relationship between TRAIL and OPG in the development and progression of OA, including how OSCAR reciprocally affects their expression.

Q6. Please show lower magnification for all IHC carried out.

Reply: Together with the answer for Q1, we now provide low magnification for all IHC images in new Figures 2h-i, 3e-g, 5e-g, Supplementary Fig. 1b, f, g, Fig. 2g-h, and Supplementary Fig. 5e-f.

Q7-1. Fig 3b-e: at which time point has been carried the experiments shown (2 or 4 weeks)

Reply: All IHC data including Fig. 3b-e (now Fig. 3d-g in the revised MS) were analyzed 8 weeks after DMM surgery. In the case of RNA seq, we originally intended to do RNA seq analysis with samples from 2-, 4-, and 7-weeks after DMM surgery. However, seven weeks after DMM surgery, we realized that the RNA amount and

quality suitable for RNA seq analysis was insufficient due to severe cartilage degradation. Therefore, we combined articular cartilage from ten mice for each group at 2- and 4-weeks after DMM surgery and analyzed eight different groups as shown in Fig. 3a and Supplementary Fig. 3a.

Q7-2. Figures 3 and 4: authors speculated on the functional relationship between OSCAR and TRAIL. They analysed OSCAR, TRAIL and OPG expression after collagenase treatment but did not show the same investigation in the absence of collagenase treatment.

Reply: In response to the reviewer's comment, we performed new experiments to compare OSCAR, TRAIL and OPG expression in the presence or absence of collagenase. We now demonstrate that collagenase treatment together with IL-1 β significantly increased OSCAR and TRAIL compared to that of untreated controls (Supplementary Fig. 4a, b). This point has been added to the main text (Page 9, lines 2-8) as follows:

We examined OSCAR expression in primary cultures of murine chondrocytes stimulated with interleukin-1 β (IL-1 β) or tumor necrosis factor- α (TNF- α), both of which are OA-associated pro-inflammatory cytokines³⁹. Treatment of IL-1 β , but not TNF- α , moderately increased the levels of OSCAR and TRAIL (Supplementary Fig. 4a, b). However, pretreatment with collagenase, followed by IL-1 β treatment, markedly increased levels of OSCAR and TRAIL, but not OPG, (Supplementary Fig. 4a, b) and....

Q7-3. Technical remarks: siRNA can induce off target effects and conventionally it is strongly recommended to confirm the effects with a minimum of 3 different sequences of siRNA.

Reply: We agree with the reviewer's concern. In response to the reviewer's comment, we performed new experiments with three different sequences of *Oscar* siRNAs. All of these decreased TRAIL expression and we have replaced the original Fig. 4b-e with new data. Additional *Oscar* siRNA sequences are shown in Supplementary Table 1.

Q8. OPG possesses a heparin binding domain leading to its potential binding to GAG, collagens, etc. Il-1 has been shown to upregulated OPG expression that is not found in Figure 4b. Collagenase treatment could explain this discrepancy however OPG transcript should be studied as well RANKL and LGR4. The study of TRAIL function must integrate all protagonists of the TNF/TNR superfamily members related to its regulation.

Reply: We appreciate your critical review. In response to your comment, we first examined the effects of pro-inflammatory cytokines, including IL-1 β and TNF- α , together with or without collagenase on the expression of OSCAR, TRAIL, and OPG.

As previously addressed in Q7-2, IL-1 β together with collagenase regulates OSCAR, TRAIL, and OPG expression (Supplementary Fig. 4a, b).

Second, as the reviewer pointed out, we observed that collagenase treatment downregulates the levels of OPG mRNAs. Also, this effect was counteracted by OSCAR silencing. We have replaced the previous Fig. 4b, c with new data.

Third, according to the reviewer's comment, we examined mRNA expression of RANKL and LGR4. We found that collagenase treatment together with IL-1 β induced RANKL, but not LGR4 (Author response Fig. 4). It should be noted that the ratio of OPG/RANKL is decreased in human OA chondrocytes (Kwan et al., *Rheumatology* (Oxford), 48:1482, 2009). In addition, Luo et al. (*Development*, 136:2747, 2009) reported previously that *Lgr4* mutant mice exhibited defects in osteoblast differentiation and mineralization, but not in chondrocyte proliferation and maturation. Based on this finding, we included RANKL expression data in Supplementary Fig. 4d.

Author response Fig. 4. Effects of collagenase RANKL and LGR4 expression. qRT-PCR analysis in mouse articular chondrocytes treated with IL-1 β (5 ng mL⁻¹) or TNF- α (50 ng mL⁻¹) and collagenase (50 U mL⁻¹) for 48 h ($n = 4$). Values are expressed as mean \pm SEM with two-way ANOVA. n indicates the number of biologically independent samples.

Lastly, we carried out additional experiments to determine whether treatment of IL-1 β /collagenase regulates TNFR superfamily members, such as TRAIL receptors (DR5, DcR1, and DcR2) and TNF receptors (TNFR1 and TNFR2) in chondrocytes. We found that the expression of *Tnfrsf10b* encoding DR5, but not DcR1 and DcR2, is significantly increased in chondrocytes (Supplementary Fig. 4d). These results were consistent with the RNA seq data. These points have been addressed in the revised text (Page 9, lines 6-12) as follows:

However, pretreatment with collagenase, followed by IL-1 β treatment, **markedly increased levels of OSCAR and TRAIL, but not OPG, (Supplementary Fig. 4a, b) and further induced chondrocyte apoptosis (Fig. 4a, Supplementary Fig. 4c). Furthermore, IL-1 β together with collagenase upregulated TRAIL receptor DR5 (encoded by *Tnfrsf10b*), but not DcR1 (encoded by *Tnfrsf10c*), DcR2 (encoded by *Tnfrsf10d*), or RANKL (encoded by *Tnfrsf11*) (Supplementary Fig. 4d).**

Q9. *TRAIL silencing in chondrocytes should be assessed as well as the comparison between WT and OSCAR KO mice.*

Reply: For TRAIL silencing, we have chosen a TRAIL neutralizing antibody as opposed to gene silencing. We have carried out additional experiments to examine whether blocking TRAIL affects IL-1 β -induced OSCAR and TRAIL expression. As shown in Supplementary Fig. 4e and f, TRAIL neutralization decreased OSCAR and TRAIL expression in chondrocytes in wild-type mice, but we did not observe this effect in *Oscar*^{-/-} mice. This point was described in the main text (Page 9, lines 12-15) as follows:

Interestingly, a TRAIL neutralizing antibody downregulated expression of OSCAR and TRAIL as well as cell death regulatory genes in chondrocytes from WT, but not *Oscar*^{-/-} mice, treated with IL-1 β in the presence of collagenase (Supplementary Fig. 4e, f, g).

Q10. *What was the impact of the collagenase treatment on chondrocyte adherence and viability by using microscopic approaches?*

Reply: In response to the reviewer's comment, we have carried out additional experiments to examine the effect of collagenase treatment on chondrocyte adherence and viability. We found that collagenase alone did not affect chondrocyte adherence and viability. However, collagenase together with IL-1 β significantly increased chondrocyte cell death, which was inhibited by hOSCAR-Fc. We have presented this data as Fig. 4h in the revised manuscript. This point has been added to the main text (Page 9, lines 22-23) as follows:

In addition, TUNEL staining showed hOSCAR-Fc efficiently blocks chondrocyte apoptosis (Fig. 4h).

Q11. *The investigations of cell apoptosis limited to caspase-3 must be extended to molecular networks conventionally observed in apoptosis.*

Reply: Based on RNA seq data, we performed a network analysis with cell death regulatory pathways (new Fig. 3c). We also examined the expression of apoptosis-related genes, including *Casp3*, *Casp8*, *Bax*, *Bcl2*, *Tnfrsf10b* (encoding DR5), *Tnfrsf10c* (encoding DcR1), *Tnfrsf10d* (encoding DcR2), *Tnfrsf1a* (encoding TNFR1), and *Tnfrsf1b* (encoding TNFR2) in the new Supplementary Fig. 4c, d. In addition, we have validated cell death regulatory gene expression in human OA samples (new Supplementary Fig. 3g).

Q12. *Figure 5: authors showed that OSCAR can block OA in mice. OSCAR should counterbalance its deficiency in OSCAR KO mice. This experiment should be carried out.*

Why to use 2 mg/Kg of recombinant OSCAR. How this dose has been defined?

Reply: We appreciate the reviewer’s comment for inclusion of additional data for the rescue experiment of OSCAR in *Oscar* KO mice. We believe our data provide compelling evidence for a therapeutic effect of OSCAR-Fc in OA models.

We prepared two kinds of OSCAR-Fc: mouse- and human-OSCAR-Fc and tested their therapeutic effects using two OA models: collagenase-induced OA (CIOA) and DMM surgery. Our results demonstrated that OSCAR-Fc blocks cartilage destruction caused by CIOA and DMM surgery. Further, the *in vivo* rescue experiments requires a protracted timeline: 1) It takes > two months to prepare OSCAR-expressing adenovirus. 2) Mice must be aged to 10-12 weeks prior to surgical DMM, followed by a 2 month in-life period following surgical DMM prior to harvesting joint tissues, and then > 1 month is necessary for histologic processing of joint tissues. This makes it logistically impossible for us to perform the additional experiments necessary to address all of the reviewers’ comments in the time allotted. Consequently, we now propose the need for further study of the therapeutic effects of OSCAR in OA as a strength of our study in the Discussion (Page 14, lines 15-20):

In addition, we developed OSCAR-Fc fusion proteins that inhibit pathological OA pathogenesis. Although OA, the most common form of arthritis, is the leading cause of disability, there is no effective disease-modifying therapy. Thus, the development of strategies to disrupt the OSCAR-collagen interaction with biologics, such as Fc fusion proteins and neutralizing antibodies, or small molecules may provide new therapeutics for OA.

In regard to the dose of recombinant OSCAR-Fc, we have initially performed tests with two doses: a low dose with 0.4 mg/kg and a high dose with 2 mg/kg of OSCAR-Fc. We found that the low dose of 0.4 mg/kg OSCAR-Fc had little inhibitory effects. Therefore, we used 2 mg/kg of OSCAR-Fc for all of the *in vivo* experiments (see below).

Author response Fig. 5. Low concentration of OSCAR-Fc blocks less articular cartilage degeneration in mice with induced OA. (a, b) WT mice subjected to sham or DMM surgery were IA-injected with human IgG as a control or hOSCAR-Fc (0.4 mg/kg or 2 mg/kg). Articular

cartilage sections were subjected to safranin-O staining (a) and quantitative analysis of OARSI grade (b). **(c, d)** WT mice injected with PBS or collagenase were IA-injected with human IgG or hOSCAR-Fc. Articular cartilage sections were examined by safranin-O staining (c) and scoring of OARSI grade (d).

In response to Reviewer #3's comments:

The submitted manuscript investigates the role of osteoclast-associated receptor (OSCAR) in osteoarthritis (OA). The manuscript demonstrate that OSCAR is more abundance in chondrocyte of the damaged cartilage of OA in human and mice models, that oblotion of OSCAR using knockouts or hOSCAR-Fc prevent the manifestation of OA induced by surgical destabilization of the medial meniscus (DMM) and intra-articular injection of collagenase in vivo, and that OSCAR co-stimulate IL-1 β -induced apoptotic signaling in aricualr chondrocytes. The manuscript was overall a pleasure to read, and is in my eyes novel, well presented and nearly complete.

Q1. *In the abstract, the authors state: “We identified an unexpected function of OSCAR...”. Why unexpected, bearing in mind the recent studies in RA and osteoclasts? I believe it’s very much in line with the recent studies, providing a novel extension to OA.*

Reply: We appreciate your critical review and believe that addressing these comments will improve the quality of our manuscript. In response to the reviewer’s comment, we changed the sentence as follows:

Thus, we identified a **novel** function of OSCAR as a catabolic regulator of OA pathogenesis, indicating that OSCAR blockade is a potential therapy for OA.

Q2. *In figure 1, the OSCAR immunoreactivity and gene expression is addressed in cartilage from human OA patients. Here the authors subdivide the cartilage in intact and damaged cartilage. Please clarify the definition of these regions according to the OARSI grade, as this must be variable between the five patients? Or even better scale*

the immunoreactivity according to different degrees of damage according to the OARSI grade, not just two extremes? Low magnification images of the whole cartilage and the regions investigate would help. Are the damaged regions also showing more apoptotic chondrocytes? How was the regions divided for the gene expression analysis? Note that OSCAR in the y-axis of the left panel C graph should be capital, as it's in human cartilage this is investigated.

Reply: First, we analyzed an increased number of human OA samples from n = 5 to n = 10. Furthermore, we have presented low magnification images for the immunohistochemical (IHC) studies to represent whole layers of articular cartilage. Accordingly, the previous IHC images were replaced with the new images as shown in Fig. 1c. Quantitative graphs showing OSCAR-positive cells and *Oscar* expression levels were also changed to new graphs in Fig. 1c.

Second, the age of the patient group ranges from 63 to 78 years with an OARSI grade of 6. We classified 0-2 grade as intact control (undamaged) and grade 6 as damaged samples. To rule out the effects of other underlying diseases, OA patients had no rheumatoid arthritis, metabolic disease, or other inflammatory diseases at the time of surgery. We included detailed information of enrolled patients in Source data Fig. 1c. This point has been described in the revised Methods (Page 17, lines 1-5) as follows:

Cartilage tissues with OARSI grade 6 were obtained from 10 patients with OA ranging in age from 63 to 78 years (3 males and 7 females) during total knee replacement surgery. To rule out the effects of other underlying diseases, OA patients had no rheumatoid arthritis, metabolic disease, or other inflammatory diseases at the time of surgery (Source data Fig. 1c).

Third, in response to your comments, we performed TUNEL staining to look at apoptotic chondrocytes in human OA samples. The data showed increased apoptosis in human OA chondrocytes (Supplementary Fig. 3h). We obtained undamaged (intact control) and damaged samples from each patient during total knee replacement surgery.

Lastly, we have corrected the graph in Fig. 1c.

Q3. *In figure 4, it's clear that western blots are cropped, and that several band are present outside the illustrated field. Please provide the whole western blots as supplementary data, and edit the cropping in figure 4. Gadph can be cropped much more, giving more space to Oscar, Trail and Opg. Please also present western blot with positive and negative controls in the supplementary figure (see also next point).*

Reply: We agree with the reviewer's comment. We repeated the experiments with three different sequences of *Oscar* siRNAs. Additional *Oscar* siRNA sequences are shown in Supplementary Table 1. We have replaced original Fig. 4b with a new Fig. 4c and provided the western blots in the Supplementary data.

Q4. *Validation of antibodies used for immunostaining and western blots. The authors only refer on the statement from the manufacture, which is too often fare from the truth. The reliability of the antibodies needs to be validated either with other antibodies binding other regions of OSCAR, TRAIL and OPG or by in situ hybridization. The inclusion of negative control tissues from knockout mice would provide a clear validation of the antibodies specificity.*

Reply: We appreciate your critical review. For the validation of antibodies, all antibodies were compared and analyzed in articular cartilage of WT and *Oscar*^{-/-} mice, and were confirmed through negative control experiments with IgG for each antibody. Specifically, antibodies against OSCAR, TRAIL, and OPG were analyzed by IHC for articular cartilage from *Oscar*^{-/-} KO and wild-type mice. We present these data in Fig. 1a and Fig. 3e, g. We also analyzed MMP3, MMP13, aggrecan and COL2A1 levels by IHC of articular cartilage from *Oscar*^{-/-} KO and wild-type mice (Supplementary Fig. 1f, g and Fig. 2g, h).

Q5. *In figure 5, the mice experiments only include five mice per group. This is too few, especially when bearing in mind the variability observed in the OARSI score of the DMM groups. I'm not convinced that the control versus hOSCAR-Fc is different if one don't assume that they are normally distributed, which I don't believe to be the case. Please include additional mice in each group.*

Reply: We agree with the reviewer's comment. In response to your comments, we have carried out additional experiments with ten mice per group for two OA models: The DMM model shown in Fig. 5 and the collagenase injection model shown in Supplementary Fig 5.

Q6. *The color codes of the graphs are very nice, but not completely consistent. Please use the gray/black bars consistently for all gene expression analysis and blue/red bars for all histology including the TUNEL analysis.*

Reply: We appreciate your careful review. We fixed this point in all of the Figures.

Reviewers' Comments:

Reviewer #1:

Remarks to the Author:

The authors have given the appropriate answer. I have no more questions.

Reviewer #2:

Remarks to the Author:

Authors have presented convincing and extensive revisions of their manuscript.

Reviewer #3:

Remarks to the Author:

All my concerns are address very satisfactory, and I don't have any further concerns or comments.

In response to Referees' comments:

Reviewer #1 (Remarks to the Author):

The authors have given the appropriate answer. I have no more questions.

Reply: We appreciate your positive review.

Reviewer #2 (Remarks to the Author):

Authors have presented convincing and extensive revisions of their manuscript.

Reply: We appreciate your positive review.

Reviewer #3 (Remarks to the Author):

All my concerns are address very satisfactory, and I don't have any further concerns or comments.

Reply: We appreciate your positive review.